

# What drives the latitudinal gradient in open ocean surface dissolved inorganic carbon concentration?

Yingxu Wu[1], Mathis P. Hain[1], Matthew P. Humphreys[1,2], Sue Hartman[3], Toby Tyrrell[1]

[1]University of Southampton, National Oceanography Centre Southampton, Southampton, UK
[2]Centre for Ocean and Atmospheric Sciences, School of Environmental Sciences, University of East Anglia, Norwich, UK
[3]National Oceanography Centre, Southampton, UK

*Correspondence to*: Yingxu Wu (Yingxu.wu@soton.ac.uk)

**Abstract.** Previous work has not led to a clear understanding of the causes of spatial pattern in global surface ocean DIC,
which generally increases polewards. Here, we revisit this question by investigating the drivers of observed latitudinal gradients in surface salinity-normalized DIC (nDIC) using the Global Ocean Data Analysis Project Version 2 (GLODAPv2) database. We used the database to test three different hypotheses for the driver producing the observed increase in surface nDIC from low to high latitudes. These are: (1) sea surface temperature, through its effect on the $CO_2$ system equilibrium constants, (2) salinity-related total alkalinity (TA), and (3) high latitude upwelling of DIC- and TA-rich deep waters. We find
that temperature and upwelling are the two major drivers. TA effects generally oppose the observed gradient, except where higher values are introduced in upwelled waters. Temperature-driven effects explains the majority of the surface nDIC latitudinal gradient (182 out of 223 $\mu mol\ kg^{-1}$ in the high-latitude Southern Ocean). Upwelling, which has not previously been considered as a major driver, additionally drives a substantial latitudinal gradient. Its immediate impact, prior to any induced air-sea $CO_2$ exchange, is to raise Southern Ocean nDIC by 208 $\mu mol\ kg^{-1}$ above the average low latitude value. However, this
immediate effect is transitory. The long-term impact of upwelling (brought about by increasing TA), which would persist even if gas exchange were to return the surface ocean to the same $CO_2$ as without upwelling, is to increase nDIC by 74 $\mu mol\ kg^{-1}$ above the low latitude average.

## 1 Introduction

The ocean absorbs about one quarter of the anthropogenic $CO_2$ emitted every year (Le Quéré et al., 2015). It is the largest non-
geological carbon reservoir (~38000 Gt C) (Falkowski et al., 2000), containing 50 times as much carbon as the pre-industrial atmosphere, and thereby plays an important role in modulating the Earth's climate system. Approximately 97% of the oceanic carbon pool exists in the form of dissolved inorganic carbon (DIC) (Falkowski et al., 2000), which is defined as the sum of the concentrations of aqueous $CO_2$, bicarbonate and carbonate ions (Zeebe and Wolf-Gladrow, 2001):

$$DIC = \left[CO_2^*\right] + \left[HCO_3^-\right] + \left[CO_3^{2-}\right] \tag{1}$$

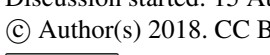



Understanding the distribution of DIC in seawater is essential for understanding anthropogenic $CO_2$ invasion (Gruber, 1998; Humphreys et al., 2016; Lee et al., 2003; Sabine et al., 1999; Sabine et al., 2002; Vázquez-Rodríguez et al., 2009) and ocean acidification (Doney et al., 2009; Orr et al., 2005). Given that the surface ocean is where most organisms live, and that it is the part of the ocean that exchanges $CO_2$ with the atmosphere, the controls on surface DIC particularly merit investigation. This

is the topic that we address in this paper, giving particular attention to the question of what sustains the high DIC concentrations observed in the Southern Ocean, given that region's crucial role in the global overturning circulation (e.g., Marshall and Speer, 2012), deep water formation (e.g., Orsi et al., 1999) and the global carbon cycle (Caldeira and Duffy, 2000; Landschützer et al., 2015; Marinov et al., 2006; Mikaloff-Fletcher, 2015; Takahashi et al., 2012).

Many previous studies focused on the vertical rather than the horizontal distribution of DIC, aiming to understand the relative

contributions of the different "carbon pumps" in controlling DIC variations throughout the water column (i.e., solubility pump, soft tissue pump, and carbonate pump, see details in Cameron et al., 2005; Gruber and Sarmiento, 2002; Toggweiler et al., 2003a; Toggweiler et al., 2003b). However, the solubility pump is so named because it is based on an assumption about the horizontal distribution of surface DIC – that DIC is high where new deep waters form at high latitudes, because of the effect of temperature on $CO_2$ solubility.

Previous work on the horizontal distribution of surface DIC included development of an algorithm to approximately reproduce the global surface DIC distribution (Lee et al., 2000), in which salinity-normalized DIC (nDIC) was predicted from empirically-derived functions of sea surface temperature and nitrate that varied seasonally and geographically. Key et al. (2004) depicted the global distribution of surface DIC based on an earlier version (GLODAPv1) of the dataset used here, noting that the surface DIC pattern is overall more similar to nutrients (including in the Southern Ocean, where both DIC and nutrients

are enriched) than to salinity, unlike total alkalinity (TA) whose pattern more closely resembles that of salinity (Fry et al., 2015). Using the data from the new GLODAPv2 database (Key et al., 2015; Olsen et al., 2016), surface DIC is confirmed here to have its highest values at high latitudes, like nutrients, and to reach its lowest values in the equatorial zone in each basin (Fig. 1a; more details in Sect. 3.1). Earlier studies (Lee et al., 2000; Toggweiler et al., 2003a; Williams and Follows, 2011, Sect. 6.3 "What controls DIC in the surface ocean?") suggested that temperature is of primary importance in regulating surface

DIC; under this assumption, surface waters in cool regions at high latitudes should hold more DIC than surface waters in the warm regions at low latitudes (Williams and Follows, 2011, Toggweiler et al., 2003a). Temperature may also drive latitudinal differences in carbonate ion concentration and surface aragonite saturation state ($\Omega_{arag}$) (Jiang et al., 2015; Orr et al., 2005), through its effect on $CO_2$ solubility and $CO_2$ system thermodynamics.

Unlike previous studies, Williams and Follows (2011) argued that another variable also exerts a secondary control on the

surface DIC distribution: at a given DIC, TA influences the seawater partial pressure of carbon dioxide (see also Omta et al., 2011), and therefore air-sea gas exchange. Takahashi et al. (2014) subsequently explored the seasonal climatological distributions of surface DIC using seawater $pCO_2$ from the Lamont Doherty Earth Observatory (LDEO) database and TA estimated from sea surface salinity, qualitatively attributing seasonal differences (on a regional scale) to the greater upward mixing of high-$CO_2$ deep waters in winter and summer biological carbon drawdown. Their study pointed out the great potential



of upwelling to alter surface DIC, but it was focused more on controls on DIC seasonality rather than DIC spatial variability. Since these studies, the global surface DIC database has been greatly expanded (e.g., Bates et al., 2006; Sasse et al., 2013; Takahashi et al., 2014), culminating now in GLODAPv2 (Key et al., 2015; Olsen et al., 2016), but the controls producing the global spatial DIC distribution have not been quantitatively reassessed.

Many processes are known to influence the sea surface distribution of DIC at the local scale. They can be divided overall into those which change DIC concentration by direct addition or removal (e.g., 1 and 2 of the following), and those which affect DIC indirectly (e.g., 3-5 of the following): (1) biological carbon assimilation during primary production and releasing during remineralization (Bozec et al., 2006; Clargo et al., 2015; Toggweiler et al., 2003b; Yasunaka et al., 2013); (2) transport of DIC-rich deep waters into the surface layer (Jiang et al., 2013; Lee et al., 2000); (3) seawater dilution/concentration due to
precipitation/evaporation (Friis et al., 2003); (4) warming or cooling, which alters the $CO_2$ solubility and induces air-sea gas exchange that acts to reduce air-sea $CO_2$ disequilibrium (Bozec et al., 2006; Toggweiler et al., 2003a; Williams and Follows, 2011); and (5) other processes which affect TA: if high/low TA values are not matched by high/low DIC values then the resulting low/high seawater $p$CO$_2$ stimulates ingassing/outgassing until DIC matches TA (Humphreys, 2017).

Our study differs from previous work in several ways. Firstly, whereas many previous studies looked to understand the vertical
DIC distribution, here the target is to understand the horizontal surface DIC distribution. Secondly, we look to identify the most important *processes* not just *variables* driving the surface distribution (Fig. 2). Another difference from previous studies is our use of a much larger observational global dataset – GLODAPv2.

The general consensus from previous work has been that the global pattern of sea surface temperature (SST) dominates the latitudinal distribution of surface DIC (Williams and Follows, 2011). More recently, TA has also been considered important
(Omta et al., 2011; Williams and Follows, 2011). We reassess this topic by quantitatively investigating the importance of SST to surface DIC and its primary controls (heating and cooling), and TA (evaporation and precipitation, Fry et al., 2015), and also the effect of high latitude upwelling. A novel conclusion of this study is that the latter process, whose global significance has previously been overlooked, is also very important in shaping the spatial distribution of surface ocean DIC.

We therefore evaluated the following three main hypotheses as to which processes cause the increase in surface DIC and nDIC
from low to high latitudes (Fig. 1):

(1) latitudinal variation of solar heating via its effect on sea surface temperature, and hence $CO_2$ solubility;

(2) evaporation and precipitation, through their effects on TA; and

(3) upwelling and winter entrainment through the introduction of DIC- and TA-rich deep waters to the (sub)polar surface oceans, when coupled with iron limitation of biological uptake of DIC.

It is easier to constrain the dynamics of upwelling and quantify its impact on surface DIC in the Southern Ocean (where upwelling has been more comprehensively studied, e.g., Marshall and Speer, 2012; Morrison et al., 2015) than in the subarctic North Atlantic and North Pacific Oceans (where upwards transport occurs via deep mixing in the winter, combined with upwelling in the North Pacific). Therefore, we focused on the Southern Ocean for the evaluation of the third hypothesis.





The processes that we evaluated can be separated into two categories: equilibrium and disequilibrium processes (Fig. 2). The effects of equilibrium processes (the effects through temperature and upwelled TA) change the surface ocean DIC at which air-sea $CO_2$ equilibrium would occur, and those effects can therefore persist beyond the air-sea $CO_2$ equilibrium timescale (months to year, Jones et al., 2014). The effects of disequilibrium processes (the effects through direct supply of DIC from upwelling and the effects through biological uptake of DIC in response to upwelled nutrients (principally iron, Moore et al., 2016)), on the other hand, are likely to persist only over timescales shorter than the $CO_2$ equilibrium timescale.

## 2 Methods

Data for this study were taken from GLODAPv2 (http://cdiac.ornl.gov/oceans/GLODAPv2/) (Key et al., 2015; Olsen et al., 2016). This product includes data from over 700 cruises conducted during the years 1972-2013 with a large fraction (~34%) having taken place during the period 2003-2013. These data have been subject to secondary quality control and subsequent adjustments (Key et al., 2015; Lauvset and Tanhua, 2015; Olsen et al., 2016). We define the surface ocean as waters shallower than 30 m at latitudes greater than 30°, and shallower than 20 m at latitudes less than 30° (following e.g., Fry et al., 2015; Lee et al., 2006). Only open ocean data (seafloor depth > 200 m) were included in this study (Fig. 3).

### 2.1. Data processing

We excluded regions subjected to significant perturbations from river inputs in order to remove confounding factors affecting the latitudinal distributions of DIC and nDIC on smaller length scales than being investigated here. To this end, we excluded all data from the Mediterranean Sea and Arctic Ocean (> 65° N) (Fig. 3) because they are heavily influenced by river inputs (Fry et al., 2015; Jiang et al., 2014), all data from the Red Sea because of its very high salinity (Jiang et al., 2014), as well as some data (those where S is less than 34) from other ocean areas: the Amazon River plume in the North Atlantic (5° N-10° N, > 45° W), the Ganges/Brahmaputra plume in the Bay of Bengal (> 5° N, 80-94° E) (both Fry et al., 2015) and the western North Atlantic margins (Cai et al., 2010). We also excluded relatively small low-latitude ocean areas affected by upwelling (the eastern equatorial Pacific and northern Californian upwelling regions).

Because atmospheric $CO_2$ increased during the time that the GLODAPv2 data was collected (1972-2013), DIC has also increased in response in surface ocean waters (Bates et al., 2014). To prevent DIC time trends from generating artificial spatial variability, we normalized surface DIC to a reference year of 2005. We assumed that sea surface $CO_2$ changes track atmospheric $CO_2$ changes (Feely, 2008; see also $CO_2$ Time Series in the North Pacific at https://pmel.noaa.gov/co2/file/CO2+time+series). We first calculated the change of atmospheric mole fraction of $CO_2$ ($xCO_{2,air}$) from the reference year 2005:

$$\Delta xCO_{2,air} = xCO_{2,air}^{t} - xCO_{2,air}^{2005} \tag{2}$$

where the globally averaged atmospheric $xCO_2$ data can be found at https://www.esrl.noaa.gov/gmd/ccgg/trends/.



Then we converted $\Delta xCO_{2,air}$ into $\Delta pCO_{2,air}$ (Takahashi et al., 2009) just above the sea surface, using in-situ hydrological data from sea surface water. It is then assumed that $\Delta pCO_{2,sw}$, representing the change of sea surface pCO2 relative to the year 2005, is equal to $\Delta pCO_{2,air}$.

Therefore the calculation of the sea surface $pCO_2$ normalized to year 2005 refers to:

$$pCO_{2,sw}^{2005} = pCO_{2,sw}^{t} - \Delta pCO_{2,sw} \tag{3}$$

where $pCO_{2,sw}^{t}$ was calculated from in-situ DIC, TA, temperature and salinity through $CO_2SYS$ (van Heuven et al., 2011; dissociation constants used are described in Sect. 2.3).

Since the anthropogenic $CO_2$ perturbation does not change TA, DIC normalized to 2005 was calculated with inputs of in-situ TA and $pCO_{2,sw}^{2005}$ using $CO_2SYS$ (van Heuven et al., 2011):

$$DIC^{2005} = f(T_{in\text{-}situ}, S_{in\text{-}situ}, TA_{in\text{-}situ}, pCO_{2,sw}^{2005}) \tag{4}$$

The concentration of DIC hereinafter refers to the DIC normalized to year 2005.

## 2.2 Salinity normalization

Salinity normalization was used to correct for the influence of precipitation and evaporation in the open ocean (Postma, 1964). Data were normalized to a reference salinity of 35 using a standard procedure:

$$nX = X_{obs} \times 35/S_{obs} \tag{5}$$

where nX refers to the normalized variable, $X_{obs}$ is the observed value of the variable, and $S_{obs}$ is the observed salinity.

## 2.3 Carbonate chemistry

Carbonate system variables were calculated from DIC and TA using version 1.1 of $CO_2SYS$ for MATLAB (van Heuven et al., 2011). The dissociation constants for carbonic acid and sulfate were taken from Lueker et al. (2000) and Dickson (1990),
20 respectively, and the total borate-salinity relationship from Lee et al. (2010).

## 2.4 Calculations of the effects of various processes on DIC

The methods for calculating the impacts of different processes on the surface DIC concentration are now explained in the order of hypotheses proposed, except for the second hypothesis (evaporation and precipitation through their effects on TA) which was evaluated by salinity normalization. The underlying purpose in each case is to better understand how the observed low-
25 to high- latitude gradients in (n)DIC are produced. The effect of upwelling is evaluated in the Southern Ocean, from both short- and long-term perspectives. In addition, we also quantify the effect of iron limitation, which would potentially affect the observed (n)DIC distribution.



### 2.4.1 SST-driven effect

Generally, the temperature effect on the carbonate system has two aspects. Firstly, when water temperature increases, the equilibrium between carbonate species (Eq. 6) shifts to the left, increasing the aqueous $CO_2$ and carbonate ion concentrations (Dickson and Millero, 1987):

$$CO_2 + H_2O + CO_3^{2-} \rightleftharpoons 2HCO_3^- \tag{6}$$

Secondly, $CO_2$ solubility is reduced at higher temperatures and vice versa (Weiss, 1974). Neither effect alters DIC directly but both lead to changes in $pCO_2$. A larger proportion of DIC exists as aqueous $CO_2$ at higher temperatures and the ratio of $pCO_2$ to $[CO_2]$ also increases as solubility decreases (Eq. 7, Henry's Law):

$$pCO_2 = [CO_2]/K_H \tag{7}$$

where $K_H$ is the Henry's constant (solubility) for $CO_2$, one of the $CO_2$ system equilibrium constants.

Both effects thus tend to increase sea surface $pCO_2$ as sea water warms, potentially elevating it to values above the atmospheric $pCO_2$ and thereby stimulating air-sea $CO_2$ gas exchange; the induced outgassing of $CO_2$ reduces sea surface $CO_2$ and DIC as it shifts the system towards air-sea $CO_2$ equilibrium. Therefore, for an open ocean system with gas exchange with the atmosphere, the DIC distribution has the potential to be controlled by the SST distribution, and this can by itself produce DIC

latitudinal variations.

To examine the magnitude of the expected temperature-induced DIC changes, we chose the low-latitude area as the reference, then removed the latitudinal SST variation and recalculated the open ocean surface DIC everywhere for a constant SST of 27°C (the mean sea surface temperature in the subtropics 30° S-30° N). To do this we first calculated the in-situ $pCO_2$ from observed SST, SSS (sea surface salinity), TA and DIC using $CO_2$SYS. We then altered the sea surface temperature from its

in-situ value to 27°C, which would change the solubility of $CO_2$ and induce air-sea $CO_2$ gas exchange. Then air-sea $CO_2$ gas exchange (which does not change TA) was assumed to proceed until $pCO_2$ was back to the same level as before resetting the temperature. Next, we used $CO_2$SYS to calculate $DIC_{SST=27}$ based on an input temperature of 27°C, observed salinity and TA, and the in-situ $pCO_2$ calculated as above. $DIC_{SST=27}$ thus represents temperature-normalized DIC, and should exhibit the same spatial variability as DIC except that the temperature-induced component of the variability has been removed. Finally, the

difference between observed DIC and $DIC_{SST=27}$ gives the DIC variation attributed to temperature variation:

$$\Delta DIC_{temp} = DIC_{obs} - DIC_{SST=27} \tag{8}$$

### 2.4.2. Upwelled DIC-driven effect (short-term effect of upwelling)

Upwelling of DIC-rich subsurface water is capable of increasing the surface DIC. The largest upwelling (in terms of flow rate) anywhere in the world takes place in the Southern Ocean (Talley, 2013): the upwelling there is made up of 18 Sv (Sverdrup,

1 Sv = $10^6$ m$^3$/s) of NADW (North Atlantic Deep Water), 11 Sv of IDW (Indian Deep Water), and 9 Sv of PDW (Pacific Deep Water). Subsurface waters in the Southern Ocean are considered to upwell along the neutral density isopycnals of 27.6 kg m⁻



[3], 27.9 kg m$^{-3}$ and 27.9 kg m$^{-3}$ in the southern Atlantic, Indian and Pacific Oceans, respectively (Ferrari et al., 2014; Lumpkin and Speer, 2007; Marshall and Speer, 2012; Talley, 2013).

Upwelling occurs within the ACC (Antarctic Circumpolar Current) at those latitudes where the wind stress is greatest (Morrison et al., 2015); as the upwelled water subsequently advects away, the effects are transported to nearby locations.

Therefore, instead of a direct supply from deep to surface locations such as L3, DIC is assumed to be brought to the subsurface primarily along isopycnals (shown in Fig. 4 as the black curve to L1), finally reaching the surface at L2, the zone in which upwelling occurs. Then, the upwelled subsurface water with enriched DIC, TA and nutrients feeds both branches of the overturning in the Southern Ocean. One is transported northwards via Ekman transport from L2 to L3 to join the upper branch, as shown by the black arrow towards the equator, and the other one is recycled back through the lower branch to form Antarctic

Bottom Water (AABW) (Talley, 2013). Potential effects of upwelling on sea surface temperature are not considered here, but are likely to be small because both deep water and high-latitude surface waters are cold.

We first consider the increase in DIC induced by the upwelling of deep water with high DIC concentrations. While some of the initial increase is usually removed shortly afterwards by biological export fueled by the nutrients brought up at the same time, excess DIC remains if the subsequent biological removal of DIC does not match the initial increase. Phosphorus has the

simplest nutrient behavior in the ocean with only one significant source to the ocean as a whole (river input) and one major sink (organic matter sinking to the seafloor) (Ruttenberg, 2003; Tyrrell, 1999). In this study, the salinity-normalized phosphate (nPhos) concentration was used as a proxy for calculating how much salinity-normalized DIC (nDIC) was upwelled along with it and not yet removed again by biological uptake of phosphate and DIC. We used salinity-normalized concentrations to correct for the influence of precipitation (rainfall) that dilutes DIC and phosphate concentrations in proportion to the effects

on salinity (Eq. 5). In this calculation, it was assumed that the only external source of phosphate to surface waters is from upwelling and the only subsequent loss is through export of organic matter, leading to the equation:

$$nPhos_{surf} = nPhos_{supply} - NCP/R_{C:P} \qquad (9)$$

where the subscript 'supply' indicates the end-member concentration of deep water supplied along the upwelling isopycnals (i.e., the value at L1 in Fig. 4), and the subscript 'surf' indicates the surface water concentration at some later time. NCP refers

to the total time-integrated net community production (uptake and export by biology) in carbon units, and $R_{C:P}$ is the Redfield ratio of carbon to phosphorus. $nPhos_{surf}$ refers to the observed surface value of nPhos at some location distant from where upwelling occurs.

Another possible process involved in the change of DIC during its upwelling and subsequent advection is calcium carbonate ($CaCO_3$) precipitation and dissolution (Balch et al., 2016), which alters DIC and TA with a ratio of 1:2. In order to quantify

the magnitude of this process, we used Alk$^*$ (Fry et al., 2015) as an indicator, which is capable of diagnosing $CaCO_3$ cycling in the context of the large-scale ocean circulation (see more details on Alk$^*$ distribution in Fig. 10). By attributing the change in Alk$^*$ concentrations between its supplied and surface end-members to $CaCO_3$ precipitation/dissolution, we then calculated $\Delta Alk^*_{CaCO3}$ (Eq. 11), and converted it to the change in nDIC (i.e., $0.5 \times \Delta Alk^*_{CaCO3}$).



$$Alk^*_{surf} = Alk^*_{supply} - \Delta Alk^*_{CaCO3} \qquad (10)$$

$$Alk^* = \frac{Alk_m - Alk_r + 1.36 \times NO_3^-}{S} \times 35 + Alk_r - 2300 \ \mu mol \ kg^{-1} \qquad (11)$$

where $Alk_m$ is the measured TA, $Alk_r$ is the riverine TA end-member (zero in the Southern Ocean), and 2300 μmol kg$^{-1}$ is the average TA in the low-latitude surface oceans.

Assuming the carbon source is from upwelled $CO_2$-rich deep waters and carbon sinks are from organic matter export (NCP) and $CaCO_3$ cycling, then:

$$nDIC_{surf} = nDIC_{supply} - (nPhos_{supply} - nPhos_{surf}) \times R_{C:P} - 0.5 \times \Delta Alk^*_{CaCO3} \qquad (12)$$

$R_{C:P}$ is given the standard value of 106:1 (Redfield, 1963), except for the cold nutrient-rich high-latitude region in the Southern Ocean (south of 45° S), where $R_C$ is given a lower value, of 78:1 (Martiny et al., 2013).

Three hydrographic sections, one in each of the Indian (I95E), Pacific (P150W), and Atlantic (A25W) Oceans, were used to determine the different supply concentrations ($nPhos_{supply}$, $Alk^*_{supply}$ and $nDIC_{supply}$) for each basin (see Fig. 5b inset, values here are expressed as mean ± standard error of the mean). In the Indian Ocean, $nPhos_{supply}$, $Alk^*_{supply}$ and $nDIC_{supply}$ along the 27.9 kg m$^{-3}$ isopycnal are. 2.29 ± 0.01 μmol kg$^{-1}$, 109.4 ± 1.0 μmol kg$^{-1}$, and 2273.1 ± 1.1 μmol kg$^{-1}$, respectively, as it approaches the surface. In the Pacific Ocean, $nPhos_{supply}$, $Alk^*_{supply}$ and $nDIC_{supply}$ along the 27.9 kg m$^{-3}$ isopycnal are 2.32 ± 0.01 μmol kg$^{-1}$, 108.1 ± 1.9 μmol kg$^{-1}$, and 2277.2 ± 1.8 μmol kg$^{-1}$, respectively. In the Atlantic Ocean, $nPhos_{supply}$, $Alk^*_{supply}$ and $nDIC_{supply}$ along the 27.6 kg m$^{-3}$ isopycnal are 2.28 ± 0.01 μmol kg$^{-1}$, 103.5 ± 1.1 μmol kg$^{-1}$, and 2254.6 ± 1.3 μmol kg$^{-1}$, respectively (Fig. 5a, b).

Since $nPhos_{surf}$ tends to decrease to zero upon moving northwards, due to biological uptake, $nDIC_{surf}$ has a relatively constant value in the subtropical regions (data not shown), where is not influenced by upwelling in the Southern Ocean. Because of this, the potential effect of upwelling on surface nDIC, is calculated as the excess in $nDIC_{surf}$ compared to the subtropical average value (30° S-30° N):

$$\Delta nDIC_{upw\_st} = nDIC_{surf} - \overline{nDIC_{surf} \ (30° \ S-30° \ N)} \qquad (13)$$

**2.4.3. Upwelled TA-driven effect (long-term effect of upwelling)**

Some effects of upwelling on DIC are temporary, becoming overridden later by gas-exchange. In contrast, the effect of upwelled TA persists because it changes the equilibrium DIC with respect to gas exchange ($DIC_{eq}$) (discussed also in Sect. 4.1.3). Upwelling of high-TA water has a long-lasting effect on DIC because, if all else remains constant, an increase in TA decreases the fraction of DIC that exists as $CO_2$ molecules. The resulting decrease in $CO_2$ concentration lowers seawater partial pressure of $CO_2$ ($pCO_2$), tending to make seawater $pCO_2$ lower than atmospheric $pCO_2$ which in turn drives an influx of $CO_2$ from the atmosphere, raising DIC (Humphreys et al., 2017).The effects of upwelling are complex because they consist of both direct and indirect effects on DIC (Fig. 2), lasting over both short (when DIC is altered but $DIC_{eq}$ is not) and long (when $DIC_{eq}$ is altered) timescales. The different effects and the meanings of the terms used here are illustrated in Fig. 6.



The calculation of the long-term effect of upwelling through upwelled TA in the Southern Ocean (i.e., the difference between $DIC_3$ and $DIC_0$ in Fig. 6) was achieved through five steps:

(1) calculation of TA in the Southern Ocean with the upwelling effect subtracted, $TA_{nonupw}$:

$$TA_{nonupw} = TA_{obs} - (Alk^* \times S_{obs} / 35) \qquad (14)$$

where $TA_{obs}$ is the observed in-situ TA, and $Alk^*$ is the TA tracer (Fry et al., 2015) revealing excess TA supplied by the large-scale ocean circulation (upwelling in the Southern Ocean), as well as removal by calcification and export (Eq. 12). Since $Alk^*$ is a salinity-normalized concept, it is necessary to restore it to the in-situ salinity before subtracting it from the in-situ TA.

(2) calculation of in-situ sea surface $pCO_2$, following the same method as described in Sect. 2.4.2.

(3) calculation of DIC with the effect of upwelled TA subtracted. We calculated $DIC_{nonupw}$ using CO2SYS with inputs of
$TA_{nonupw}$ and in situ $pCO_2$, SST and salinity.

(4) salinity-normalization for consistency with other calculated effects.

(5) finally, the long-term effect of upwelling through the upwelled TA and the subsequent air-sea gas exchange is calculated as:

$$\Delta nDIC_{upw\_lt} = nDIC_{obs} - nDIC_{nonupw} \qquad (15)$$

where $\Delta nDIC_{upw\_lt}$ corresponds to the magnitude of ⑤ in Fig. 6.

## 2.4.4. Iron-driven effect

The iron limitation-driven DIC differences ($\Delta DIC_{Fe}$) relate to the concepts of "unused nutrient" and associated "unused DIC", which can be thought of as the amounts of macro-nutrients and DIC that are left behind after iron limitation brings an end to biological uptake, in those regions where iron is the limiting nutrient. Iron limitation alters the impact of upwelling. In locations
experiencing upwelling but where nitrate is the proximate limiting nutrient, then the quantity of upwelled DIC might more or less balanced by the quantity of subsequently exported DIC (fueled by the upwelled nitrate). In the Southern Ocean, however, the two appear not to be close to balance, even before considering iron limitation. According to the calculations in Sect. 2.4.3, the ratio of upwelled nDIC against nPhos is around $250:2.3 \approx 109:1$ for the Southern Ocean, considerably exceeding the low C:P (average $\approx 80:1$) of organic matter in the region (Martiny et al, 2013). So even if all upwelled phosphate were to be used
up and then exported in biomass in conjunction with carbon, a considerable surplus of DIC would be left behind. A lack of iron in surface waters, however, leads to even more upwelled DIC being left behind after the end of blooms induced by the upwelled nutrients.

Because phosphorus has greater plasticity than nitrogen in plankton stoichiometry, P:C (i.e. the elemental ratio of phosphorus:carbon) exhibits significantly more variability in the ocean than does N:C (Galbraith and Martiny, 2015). Because
of this, we chose in this instance to use nitrate as the "unused nutrient" from which to calculated "unused DIC".

For each $1° \times 1°$ grid in the surface open ocean, the unused nitrate was taken from its annual minimum concentration based on the monthly data in World Ocean Atlas 2013 version 2 (WOA 2013: https://www.nodc.noaa.gov/OC5/woa13/, Boyer et al.,



2013). The unused nitrate was then converted into unused DIC based on a C:N ratio of 106:16 (Redfield, 1963) for most of the global ocean, except in the warm nutrient-depleted low-latitude gyres, warm nutrient-rich equatorial upwelling regions, and cold nutrient-rich high-latitude regions. The C:N ratios used for these three regions were 195:28, 137:18, and 78:13, respectively (Martiny et al., 2013).

The amount of unused DIC was therefore calculated as:

$$\text{unused DIC} = \text{unused nitrate} \times R_{C:N} \tag{16}$$

## 2.5. Uncertainty estimation

In this study, uncertainties in calculated effects of different drivers (e.g., $\Delta nDIC_{temp}$, $\Delta nDIC_{temp-upw}$, $\Delta nDIC_{upw\_st}$, $\Delta nDIC_{upw\_lt}$) were determined by a Monte Carlo approach. For example, the uncertainty of $\Delta nDIC_{temp}$ was calculated as follows: (1) given

that $\Delta nDIC_{temp}$ is the difference between $nDIC_{obs}$ and $nDIC_{SST=27}$ (Eq. 9), its uncertainty is propagated from the uncertainties of both $nDIC_{obs}$ and $nDIC_{SST=27}$, where the uncertainty of $nDIC_{obs}$ is 5 µmol kg$^{-1}$ (Table 2), and the uncertainty of $nDIC_{SST=27}$ was determined by a Monte Carlo approach; (2) for calculation of the uncertainty of $nDIC_{SST=27}$ (see its function in Table 1), we first calculated artificial random errors (normally distributed according to the central limit theorem, with a mean of zero and a standard deviation equal to the accuracy/uncertainty of measurement) using a random number generator. Then, new

carbonate system variable values (the original ones plus the randomly generated errors) were input into the CO$_2$SYS program (Van Heuven et al., 2011) to calculate new $nDIC_{SST=27}$ values. By doing this 1000 times, we obtained a set of 1000 different values for every single data point in the dataset. We used the standard deviations of these sets to characterize their individual uncertainties. The overall uncertainty of $nDIC_{SST=27}$ was 6.4 µmol kg$^{-1}$; (3) by applying the same Monte Carlo method, but to calculate the uncertainty propagated through Eq. (9), we then calculated the uncertainty of $\Delta nDIC_{temp}$ to be 8.0 µmol kg$^{-1}$

(Table 2).

This Monte Carlo approach has been used previously (e.g., Juranek et al., 2009; Ribas-Ribas et al., 2014) to propagate uncertainties involving CO$_2$ system calculations.

## 3. Results

### 3.1. Spatial distributions of DIC and nDIC

Surface observations reveal values of DIC across the global ocean ranging from less than 1850 µmol kg$^{-1}$ in the tropics to more than 2200 µmol kg$^{-1}$ in the high latitudes (Fig. 1a). To first order, surface DIC increases polewards, being positively correlated with absolute latitude (Spearman's rank correlation coefficient $\rho = 0.71$ for the global oceans, Table 3). Spatially, it is monotonically inversely related to sea surface temperature ($\rho = -0.78$, Table 3), with DIC being highest where the surface ocean is coolest. Another conspicuous feature of surface DIC is the higher values (~100 µmol kg$^{-1}$ higher) in the tropical and

subtropical Atlantic Ocean relative to the same latitudes in the Pacific and Indian Oceans (Fig. 1a), due to the intense





evaporation in the subtropical Atlantic Ocean, as well as the transport of water vapor from the Atlantic to the Pacific (Broecker, 1989). This is not considered further here because our main purpose is to explain the sizeable observed latitudinal gradients in DIC (on average 153 µmol kg$^{-1}$ higher in the Southern Ocean than at low latitudes, for instance) and nDIC (on average 223 µmol kg$^{-1}$ higher in the Southern Ocean than at low latitudes).

Salinity-normalized DIC (nDIC) increases towards the poles in all three ocean basins (Fig. 1b), although less strongly in the North Atlantic. The surface nDIC correlates more tightly with latitude and SST than does DIC, yielding a positive correlation with absolute latitude and a negative correlation with SST ($\rho = 0.86$ and $-0.94$ respectively for the global ocean, Table 3). The distributions of surface DIC and particularly nDIC also show modest regional maxima in the eastern equatorial Pacific, the Arabian Sea, and the eastern boundaries of the Pacific and Atlantic Ocean basins, presumably as a result of upwelling

(Capone and Hutchins, 2013; Chavez and Messié, 2009; Millero et al., 1998; Murray et al., 1994).

### 3.2. SST-driven effect in the global surface ocean

The differences between the latitudinal patterns of DIC$_{obs}$ and DIC$_{SST=27}$ are shown in Fig. 7. As expected, DIC$_{SST=27}$ agrees well with DIC$_{obs}$ in the subtropics where SST is close to 27°C; the differences become larger with increasing latitude and decreasing SST (Fig. 7a-c). Correcting for salinity variations (Fig. 7d-f) greatly reduces the variability in DIC at low latitudes:

nDIC$_{obs}$ is fairly constant at ~1970 µmol kg$^{-1}$ in the subtropics. $\Delta$nDIC$_{temp}$, the temperature-driven CO$_2$ gas exchange effect on surface nDIC, increases sharply with latitude (Fig. 7g-i), reaching ~220 µmol kg$^{-1}$ at 70° S, and ~200 µmol kg$^{-1}$ at 60° N in the northern part of the Atlantic and Pacific Oceans, with an average value of 182 µmol kg$^{-1}$ in the Southern Ocean, which is large enough to account by itself for most - but not all - of the nDIC latitudinal gradient of 223 µmol kg$^{-1}$ (2193-1970 µmol kg$^{-1}$). The estimated overall uncertainty of SST-driven effect on surface nDIC (Table 2) ranges from 5 to 8 µmol kg$^{-1}$, which is of

comparable magnitude to the uncertainty of DIC normalized to 2005, and much smaller than the large latitudinal variations of $\Delta$nDIC$_{temp}$.

### 3.3. Upwelling-driven effect in the Southern Ocean

The upwelling-driven effects in the Southern Ocean calculated from both short- and long-term perspectives are shown in Fig. 8.

$\Delta$nDIC$_{upw\_st}$ increases polewards (Fig. 8a-c), with the same trends as surface phosphate (not shown), because it is calculated from phosphate. It can be seen that surface nDIC is potentially elevated dramatically by the Southern Ocean upwelling (up to about 250 µmol kg$^{-1}$). The effect is of larger magnitude (average of 208 µmol kg$^{-1}$ in the Southern Ocean) than that calculated for $\Delta$nDIC$_{temp}$ (Fig. 7g-i). Fig. 8d-f show the long-term effect of upwelling, which are controlled by the concentration of TA in the upwelled water (how

much upwelling increases surface TA values by). The average magnitude of $\Delta$nDIC$_{upw\_lt}$ is around 74 µmol kg$^{-1}$ for the Southern Ocean.





The estimated overall uncertainty of upwelling-driven effects on surface nDIC (Table 2) ranges from 5 to 9 µmol kg$^{-1}$, close to the uncertainty of DIC normalized to 2005, and much smaller than the large latitudinal variations of $\Delta$nDIC$_{upw\_st}$ and $\Delta$nDIC$_{upw\_lt}$.

### 3.4. Iron-driven effect in the global surface ocean

As shown in Fig. 9, $\Delta$DIC$_{Fe}$ is close to zero except in the classic HNLC regions (i.e., the North Pacific, the equatorial Pacific, and the Southern Ocean, Moore et al., 2013). There is also some residual nitrate during most summers in the Iceland and Irminger Basins of the North Atlantic due to the seasonal iron limitation there (Nielsdóttir et al., 2009). The surface Southern Ocean south of 40° S has the largest unused DIC ($\Delta$DIC$_{Fe}$ of up to 165 µmol kg$^{-1}$, average of 95 µmol kg$^{-1}$), followed by the North Pacific 40° N-65° N ($\Delta$DIC$_{Fe}$ of up to 90 µmol kg$^{-1}$, average of 30 µmol kg$^{-1}$) and the equatorial Pacific (average of 18

µmol kg$^{-1}$). It is negligible elsewhere in the tropics and subtropics.

### 4. Discussion

### 4.1. Factors controlling the surface DIC latitudinal variation

As shown in Fig. 7d-f, nDIC is about 223 µmol kg$^{-1}$ higher in the Southern Ocean, and about 192 µmol kg$^{-1}$ higher in the North Pacific, than at low latitudes (Table 4). Smaller increases of DIC are calculated, of about 150 µmol kg$^{-1}$ in the Southern Ocean,

50 µmol kg$^{-1}$ in the North Pacific and 100 µmol kg$^{-1}$ in the North Atlantic (Fig. 7a-c). Both DIC and nDIC are clearly higher, on average, at high latitudes. Fig 7d-f also show nDIC residuals after the estimated temperature effect is removed (nDIC$_{SST=27}$), of around 50 µmol kg$^{-1}$ polewards in all basins in the southern hemisphere and of around 60 µmol kg$^{-1}$ (even higher) in the North Pacific. The next few sections consider the contributions of different individual processes to the elevation of DIC and nDIC at high latitudes. The observed gradients are not solely due to temperature.

**4.1.1. Effect of SST variation in the global surface ocean**

The previously accepted explanation for higher DIC at high latitudes is that cooler SSTs there increase CO$_2$ solubility, resulting in a higher equilibrium DIC (Toggweiler et al., 2003a; Williams and Follows, 2011). Our results support an important role for SST, but also that other processes contribute significantly.

Our analysis concludes that the latitudinal gradient in temperature is capable of raising nDIC by about 180 µmol kg$^{-1}$ in the

Southern Ocean, or in other words of explaining about four-fifths of the observed gradient of 223 µmol kg$^{-1}$. SST variation is thus able to explain most of the observed pattern.





### 4.1.2. Effect of TA distribution in the global surface ocean

A second factor that has been proposed as influential in driving spatial variations in the concentration of DIC in the surface ocean is TA (Williams and Follows, 2011). Our analysis supports this contention, although we note that the effect of TA is most prominent at low latitudes. Large differences in DIC are observed between the subtropical gyres, where values are

relatively high, and the vicinity of the equator, where values are relatively low (Fig. 1a). These differences are driven by the effects of evaporation and precipitation on TA, which then drive differences in DIC because of the influence of TA on DIC at gas exchange equilibrium with a given atmospheric $CO_2$ level. The role of TA explains the much clearer relationship between latitude and nDIC than between latitude and DIC (Fig. 1, Table 3); normalizing DIC to salinity is almost the same as normalizing DIC to TA, because salinity and TA are highly correlated in the surface ocean. As a result, the effect of TA on

DIC is counteracted by salinity normalization, with the pattern in nDIC (Fig. 1b) then revealing more clearly how other factors impact DIC.

The latitudinal pattern in TA is not the dominant driver of the DIC trend, because TA values are generally lower at high latitudes (where precipitation often exceeds evaporation) than they are at low latitudes (where evaporation often exceeds precipitation). However, TA is also biologically cycled and thus not perfectly correlated to salinity (Fry et al., 2015) and the

presence of excess TA in deep water upwelled at high latitudes does contribute to the DIC trend.

### 4.1.3. Effect of upwelling in the Southern Ocean

Although not traditionally considered as a factor, our analyses show that upwelling is important in driving the latitudinal gradient in DIC. Upwelling of DIC by itself is capable of producing an nDIC latitudinal gradient of 208 μmol kg$^{-1}$ in the Southern Ocean, even higher than the effect of temperature (Fig. 8a-c, Table 4). However, the contribution of upwelling is

reduced by about two thirds if only the long-term effect through upwelled TA is considered (see Fig. 6 for definitions of terms). Deep water usually has higher concentrations of nutrients, DIC and nTA than does surface water. Introduction of deep water into the surface mixed layer therefore usually stimulates increases in these concentrations, with three main consequences for DIC (Fig. 6), as follows. (A) If the upwelled water has higher DIC than the surface then the upwelling causes an immediate initial increase in DIC; (B) additional nutrients stimulate phytoplankton blooms until the proximate limiting nutrient runs out,

leading to a reduction in DIC over timescales of days to weeks (or months if, for instance, the upwelling occurs at high latitudes during winter when phytoplankton cannot bloom); (C) finally, air-sea gas exchange tends to remove any upwelling-induced air-sea $CO_2$ disequilibrium over a period of months to a year (Jones et al., 2014), although the full equilibrium is seldom achieved across the global surface ocean (Takahashi et al., 2014). However, the amount of ingassing/outgassing potentially required to restore air-sea $CO_2$ equilibrium is a function of the amount of upwelled TA (which, together with temperature,

controls the equilibrium DIC).

The upwelling effects in Fig. 8 are calculations based on phosphate and TA concentrations, taking into account both the amount upwelled, and the amount subsequently removed by biology. They therefore correspond to the sum of the direct upwelling



effect (①  in Fig. 6) and the indirect upwelling effect through supplied nutrients (② in Fig. 6). There are two reasons why the initial amount of upwelled DIC considerably exceeds the amount of DIC subsequently taken up by phytoplankton growth fueled by the upwelled nutrients (why ① > ②) in the Southern Ocean.

Firstly, iron is typically much scarcer in deep waters than are macronutrients, relative to phytoplankton need (Moore, 2016).

Regions like the Southern Ocean that are strongly influenced by upwelling are for this reason often iron-limited (Moore, 2016), leading to large amounts of 'unused DIC' (order of 95 µmol kg$^{-1}$ in the Southern Ocean - Fig. 9) accompanying unused macronutrients. This scarcity of iron also leads to muted seasonal cycles of DIC (Merlivat et al., 2015) and thus year-round persistence of unused DIC. Secondly, as described in Sect. 2.4.5, the higher C:P ratio of supply (~109:1) compared to removal (~80:1) implies a considerable surplus of DIC even without iron limitation.

The upwelling effects shown in Fig. 8a-c are however relatively short-term, and are expected to be overridden by air-sea gas exchange within months (Jones et al., 2014). They are thus likely to be most significant in the vicinity of where upwelling takes place (Morrison et al., 2015). For effects that may persist further away from locations of upwelling, it is important to consider also the long-term effect (⑤ in Fig. 6), the magnitude of which is dictated mainly by the change in TA brought about by upwelling. The level of TA in upwelled water (~2315, 2340, and 2337 µmol kg$^{-1}$ in the Atlantic, Indian and Pacific sectors

of Southern Ocean, respectively; calculated according to the same method as in Sect. 2.4.3) are higher than the typical levels of TA in the surface waters of the high latitude Southern Ocean (~2300, 2289, and 2288 µmol kg$^{-1}$ in the Atlantic, Indian and Pacific sectors, respectively). The increase in TA brought about by upwelling corresponds to a long-term upwelling effect on nDIC of about 74 µmol kg$^{-1}$ (Fig. 8d-f) in the Southern Ocean.

Our results show that upwelling in the Southern Ocean can, by itself, generate high-latitude nDIC values that are around 208

µmol kg$^{-1}$ greater than subtropical values. We emphasize that there is, in addition, a sizeable long-term effect of upwelling (forcing nDIC values to be around 74 µmol kg$^{-1}$ higher than they would be otherwise). Contrary to what might typically be assumed, the long-term effects of upwelling are dictated by the amounts of TA upwelled, and not by the amounts of DIC or nutrients.

### 4.2. A new understanding of the controls on the surface DIC distribution

The analyses presented above show that our understanding of the causes of latitudinal gradient in DIC needs to be revised. Whereas it was considered previously that the latitudinal gradient in DIC is completely explained by sea surface temperature variation, here we have shown that upwelling is also an important contributor to it, based on an evaluation in the Southern Ocean. DIC and nDIC would still be elevated at high latitudes even if there was no temperature effect (even if $CO_2$ solubility was completely unaffected by SST). The incomplete previous view of what drives the DIC latitudinal gradients should be

replaced by this more nuanced view in which upwelling is also seen to contribute.

It seems that neither temperature variation nor upwelling are responsible for all of the observed large latitudinal gradients in DIC and nDIC (for instance, the ~223 µmol kg$^{-1}$ difference in nDIC between low-latitudes and the Southern Ocean), but rather that they are jointly responsible. There is an apparent contradiction because both $\Delta nDIC_{temp}$ and $\Delta nDIC_{upw\_st}$ appear to account





for more than 80% of the nDIC latitudinal gradient. While both processes are capable individually of raising DIC by 182 and 208 µmol kg⁻¹ in the Southern Ocean, acting together they raise it by only 223 µmol kg⁻¹ instead of 390 µmol kg⁻¹. An obvious explanation of this apparent paradox is that when we consider upwelling effects, we should consider not only its short-term effect through supplying DIC and nutrients (① + ② in Fig. 6), but also its long-term effect with gas exchange with the
atmosphere in the context of elevated TA by upwelling (⑤ = ① + ② + ③ in Fig. 6). The sum of the SST-driven effect and the long-term effect of upwelling approximately equals the nDIC latitudinal gradient (Table 4).

On the global scale, therefore, the ultimate controls on the surface DIC and nDIC latitudinal gradients are the spatial patterns of SST and upwelling, and the chemical composition of the upwelled water.

### 4.3. Importance of upwelling confirmed by the North Atlantic

From inspection of the global nDIC distribution (Fig. 1b), it can be seen that nDIC increases with latitude in all basins, but, as shown in Table 4, does so less strongly in the North Atlantic (difference between high latitudes and low latitudes of 114 µmol kg⁻¹) than in the North Pacific (difference of 192 µmol kg⁻¹). Although the latitudinal temperature gradient is less pronounced in the North Atlantic, this is not enough to explain the variation in gradients between the two basins: the average temperature of the high-latitude North Atlantic is 12.4°C and of the high-latitude North Pacific is 9.5°C, which can explain about 20 µmol
kg⁻¹ of variation between the two nDIC gradients but cannot explain the observed 78 µmol kg⁻¹ variation (Table 4).

The reason for the discrepancy is that the Southern Ocean and the North Pacific experience elevations in values due to inputs of deep water whereas the North Atlantic does not. Upwelling occurs in the Southern Ocean and entrainment due to deep winter mixing occurs in the subarctic North Pacific (Mecking et al., 2008; Ohno et al., 2009) where it entrains waters high in both TA (Fry et al., 2016) and DIC. While deep winter mixing also occurs in the high latitude North Atlantic (de Boyer
Montégut et al., 2004), the entrained waters left the surface relatively recently and hence there is little accumulated remineralized DIC and TA in the deep water that is reintroduced to the surface. For this reason, winter entrainment produces little increase in surface nDIC in the North Atlantic. This makes the North Atlantic useful in discriminating between the two effects because, uniquely out of the three regions, only the SST effect operates there. As expected, the SST effect is able to completely account for the observed nDIC gradient in the North Atlantic, whereas it cannot in the other two regions (columns
2 and 3 of Table 4). The North Atlantic confirms the important contribution of upwelling to latitudinal gradients, while also showing that latitudinal gradients occur in the absence of upwelling.

### 4.4. Comparison of nDIC distribution to Alk* and nutrients

Fig. 10 shows a comparison between the patterns of nDIC, the TA tracer Alk* (Eq. 11, Fry et al., 2015) and salinity-normalized nutrients. The similarities and differences in distributions of Alk* and nutrients have previously been discussed by Fry et al.
(2015). Here we extend the comparison to also include nDIC. All exhibit low and fairly constant values at low latitudes, primarily due to biological uptake and restricted supply from subsurface waters, for most variables, but primarily due to fairly uniform high temperatures for nDIC. All increase polewards due to upwelling/entrainment (also SST for nDIC), exhibiting





maxima at high latitudes in the Southern Ocean and North Pacific. All exhibit a more modest increase in the North Atlantic than in the North Pacific, because the deep water formed relatively recently. The coincident increases in nDIC and nitrate in the north Indian Ocean and equatorial Pacific Ocean are not matched by increases in either Alk[*] or silicate, most probably due to the source waters for the upwelling coming from depths that are shallower than the dissolution depths of calcium carbonate

and opal (Fry et al., 2015; Schlitzer, 2000). There are differences in the latitudes at which the different parameters start to increase on a transect from the equator towards Antarctica, reflecting the different processes involved. Surface nDIC is the first to start increasing, under the influence of SST (rows 3 and 4 of Fig. 7), at around 20° S in the Atlantic and Pacific Oceans and 25° S in the Indian Ocean. Alk[*] and nitrate, on the other hand, do not start to increase until about 40° S in the Atlantic and Indian Oceans and about 30° S in the Pacific Ocean. Silicate does not increase in concentration until about 50° S, for reasons

that are still debated (Assmy et al., 2013; Holzer et al., 2014; Vance et al., 2017).

**4.5. Implications for the future CO₂ sink under climate change**

It is widely understood that global warming may alter the spatial distribution and intensity of upwelling in the ocean (Bakun, 1990; McGregor et al., 2007; Wang et al., 2015). It could either increase it on average, due to higher average wind speeds in a warmer, more energetic atmosphere (Bakun, 1990; Wang et al., 2015), or decrease it on average, due to enhanced stratification

as the temperature differential between surface and deep waters is increased (Barton et al., 2013; Sarmiento et al., 2004b). Furthermore, it is widely understood that an increase in upwelling would lead to an increase in the amount of CO₂ outgassed from the ocean, as larger quantities of CO₂-rich deep water are brought up to the surface and their CO₂ vented to the atmosphere (Evans et al., 2015; Marinov et al., 2006; Morrison et al., 2015). However, we have identified an additional effect here. Changes in upwelling would alter the distribution of carbon in the surface ocean not only through the supply of CO₂, but also through

the supply of TA which determines the eventual surface carbonate system equilibrium with the same atmospheric $p$CO₂ (Humphreys et al., 2018). That is to say, the impact of changes in upwelling on the ocean's carbon source/sink strength depends not only on the DIC content of the upwelled water but also on its TA content. Ocean carbon cycle models should include these additional consequences if they are to make accurate predictions about the impacts of global warming on future carbon cycling. They should include the several routes identified here by which upwelling affects surface DIC: through upwelling of DIC,

through upwelling of nutrients, and through upwelling of TA.

**5. Conclusions**

We investigated the global surface DIC and nDIC distributions in order to explain the large differences between high latitude (especially Southern Ocean) and low-latitude regions. This issue has been addressed in previous studies and here we revisited it using new analytical approaches that lead to new findings. We considered three drivers for how the phenomenon could be

explained: (1) sea surface temperature variations through their effect on CO₂ system equilibrium constants, (2) salinity-related



TA variations through their effect on $p$CO$_2$, and (3) upwelling in the subpolar oceans. Our analyses confirmed that temperature plays a dominant role through its effect on solubility, and is able to explain a large fraction of the surface nDIC latitudinal gradient (182 μmol kg$^{-1}$ out of 223 μmol kg$^{-1}$ in the high-latitude Southern Ocean). Variations in TA associated with evaporation and precipitation are unable to explain higher DIC concentrations at higher latitudes, because alone they would drive the opposite DIC pattern. Their role is therefore to reduce the magnitude of the polewards gradient in DIC. Upwelling, whose role in driving the large-scale spatial patterns has not previously been appreciated, accounts for a sizeable component of the surface nDIC latitudinal gradient (up to 250 μmol kg$^{-1}$, on average 208 μmol kg$^{-1}$ in the Southern Ocean). Its importance is magnified by the iron limitation that frequently occurs in upwelling areas, leaving behind residual upwelled excess DIC and macronutrients that cannot be utilized by biology. We emphasize that the upwelling of TA alongside DIC generates a prolonged effect that persists beyond CO$_2$ gas exchange re-equilibration timescales. The long-term effect of upwelling (74 μmol kg$^{-1}$ in the Southern Ocean) helps explain the shortfall between the observed nDIC latitudinal gradient (223 μmol kg$^{-1}$) and the magnitude of the temperature-driven effect (182 μmol kg$^{-1}$). On the global scale, we conclude that no single mechanism accounts for the full amplitude of surface DIC latitudinal variations but that temperature and the long-term effect of upwelling, in that order, are the two major drivers.

## Acknowledgements

Data for this study came from the Global Ocean Data Analysis Project Version 2 (GLODAPv2) and World Ocean Atlas 2013 version 2 (WOA13 V2). All the data used is publicly available at the Carbon Dioxide Information Analyses Center (CDIAC, http://cdiac.ornl.gov/oceans/GLODAPv2/) and National Oceanic and Atmospheric Administration (NOAA, https://www.nodc.noaa.gov/OC5/woa13/woa13data.html). This study was funded by the Swire Educational Trust (PhD studentship to Yingxu Wu), and we also acknowledge funding by RAGNARoCC to M. P. Humphreys and T. Tyrrell (NE/K002546/1).

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

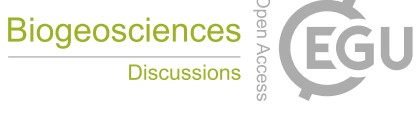

**Table 1**

**Definitions of Subscripts and Main Terms used in the Text.** X represents any variable involved in the calculations. The program $CO_2SYS$ was used to calculate values under different conditions.

| Subscript | Meaning |
| --- | --- |
| *Referring to (n)DIC values at a particular location* | |
| $X_{supply}$ | Value at depth, along isopycnals that upwell at this location |
| $X_{surf}$ | Predicted value in the surface layer |
| $X_{obs}$ | Observed value at this surface location |
| *Referring to predicted (n)DIC values under different conditions* | |
| $X_{SST=27}$ | Predicted value with sea surface temperature changed to 27°C |
| $X_{nonupw}$ | Predicted value with upwelled TA subtracted |
| *Referring to changes in (n)DIC values because of processes* | |
| $\Delta X_{Fe}$ | Effect of iron limitation (biological drawdown that is prevented) |
| $\Delta X_{temp}$ | Effect of sea surface temperature variations |
| $\Delta X_{upw\_st}$ | Short-term effect of upwelling, through upwelled DIC |
| $\Delta X_{upw\_lt}$ | Long-term effect of upwelling, through upwelled TA |
| *Carbonate variables used to calculate predicted DIC values with $CO_2SYS$* | |
| $DIC_{SST=27} = f(T_{SST=27}, S_{in\text{-}situ}, TA_{in\text{-}situ}, pCO_{2,in\text{-}situ})$ | $DIC_{SST=27}$ is a function of in-situ S, TA, and $pCO_2$, and SST at 27°C |
| $DIC_{nonupw} = f(T_{in\text{-}situ}, S_{in\text{-}situ}, TA_{nonupw}, pCO_{2,in\text{-}situ})$ | $DIC_{nonupw}$ is a function of in-situ SST, S, and $pCO_2$, and pre-upwelling TA |





**Table 2.**

**Uncertainties for variables in this study.**

| Initial Variable | Uncertainty | Reference |
|---|---|---|
| Salinity | 0.005 | Olsen et al. (2016) |
| Phosphate | 0.05 μmol kg$^{-1}$ | Olsen et al. (2016) |
| DIC | 4 μmol kg$^{-1}$ | Olsen et al. (2016) |
| TA | 6 μmol kg$^{-1}$ | Olsen et al. (2016) |
| $p$CO$_2$ | 6.8 μatm | Takahashi et al. (2014) |
| DIC normalized to 2005 | 5.0 μmol kg$^{-1}$ | derived in this study[a] |
| Alk$^*$ | 6.1 μmol kg$^{-1}$ | Modified from Fry et al. (2015) |
| nPhos$_{supply}$, Alk$^*_{supply}$, nDIC$_{supply}$ | See text in Sect. 2.4.3 | derived in this study[b] |
| *Calculated Propagated Uncertainties* | | |
| ΔnDIC$_{temp}$ | 8.0 μmol kg$^{-1}$ | derived in this study |
| ΔnDIC$_{upw\_st}$ | 5.4 μmol kg$^{-1}$ | derived in this study |
| ΔnDIC$_{upw\_lt}$ | 8.9 μmol kg$^{-1}$ | derived in this study |

[a]the uncertainty of DIC normalized to 2005 was primarily propagated from TA and $p$CO$_{2,sw}^{2005}$. The uncertainty of $p$CO$_{2,sw}^{2005}$ was calculated from error propagation (Fornasini, 2008), to be 0.17 μatm.

5    [b]the uncertainties for variables with subscript "baseline" and "supply" were from their standard error of the mean.





**Table 3**

**Global and Regional Correlations Between DIC, nDIC and SST, Latitude.**

| Ocean | Region | DIC vs. Lat | | nDIC vs. Lat | | DIC vs. SST | | nDIC vs. SST | |
|---|---|---|---|---|---|---|---|---|---|
| | | $\rho$[a] | N[b] | $\rho$ | N | $\rho$ | N | $\rho$ | N |
| Global | | **0.71** | 14228 | **0.86** | 14228 | **-0.78** | 14228 | **-0.94** | 14228 |
| Southern Ocean | S of 40° S | **0.79** | 3061 | **0.81** | 3061 | **-0.93** | 3061 | **-0.95** | 3061 |
| North Atlantic | N of 40° N | **0.30** | 1640 | **0.58** | 1640 | **-0.34** | 1640 | **-0.78** | 1640 |
| North Pacific | N of 40° N | **0.02** | 1601 | **0.34** | 1601 | **-0.78** | 1601 | **-0.87** | 1601 |

[a]the Spearman's rank correlation coefficient, for assessing monotonic relationships (there is a non-linear relationship between SST and $CO_2$ solubility). Statistically significant correlations are shown in bold.

5  [b]The number of data points from that area that were used in calculating the correlations.





**Table 4.**

**Summary of nDIC Differences Between Low and High Latitudes.** Each $\Delta$nDIC value is the amount by which the average nDIC value for the high latitude region exceeds the average value for the low latitudes (30° S to 30° N). %'s in brackets represent the ratio to the observed nDIC difference in column 2. n.c. = not calculated.

| Region[a] | Observed $\Delta$nDIC ($\mu$mol kg$^{-1}$) | $\Delta$nDIC$_{temp}$ ($\mu$mol kg$^{-1}$) | $\Delta$nDIC$_{upw\_st}$ ($\mu$mol kg$^{-1}$) | $\Delta$nDIC$_{upw\_lt}$ ($\mu$mol kg$^{-1}$) |
|---|---|---|---|---|
| Southern Ocean | 223 | 182 (82%) | 208 (93%) | 74 (33%) |
| North Atlantic | 114 | 122 (107%) | n.c. | n.c. |
| North Pacific | 192 | 137 (71%) | n.c. | n.c. |

5 [a]The regions are defined as follows: North Atlantic: 40° N - 60° N; North Pacific: 40° N - 60° N; Southern Ocean: S of 40° S.




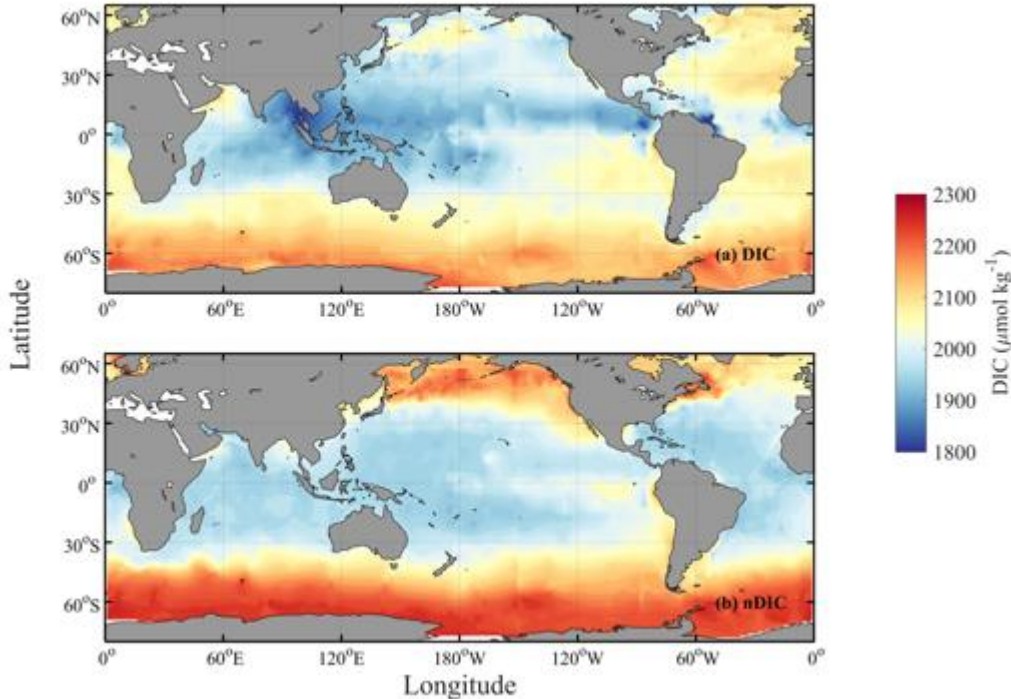

**Figure 1. Spatial distributions of DIC and nDIC.** (a) DIC (normalized to year 2005), (b) salinity-normalized DIC (nDIC, DIC normalized

to reference year of 2005 and salinity of 35) in the surface global ocean. The latitudinal trends are clear, particularly for nDIC.





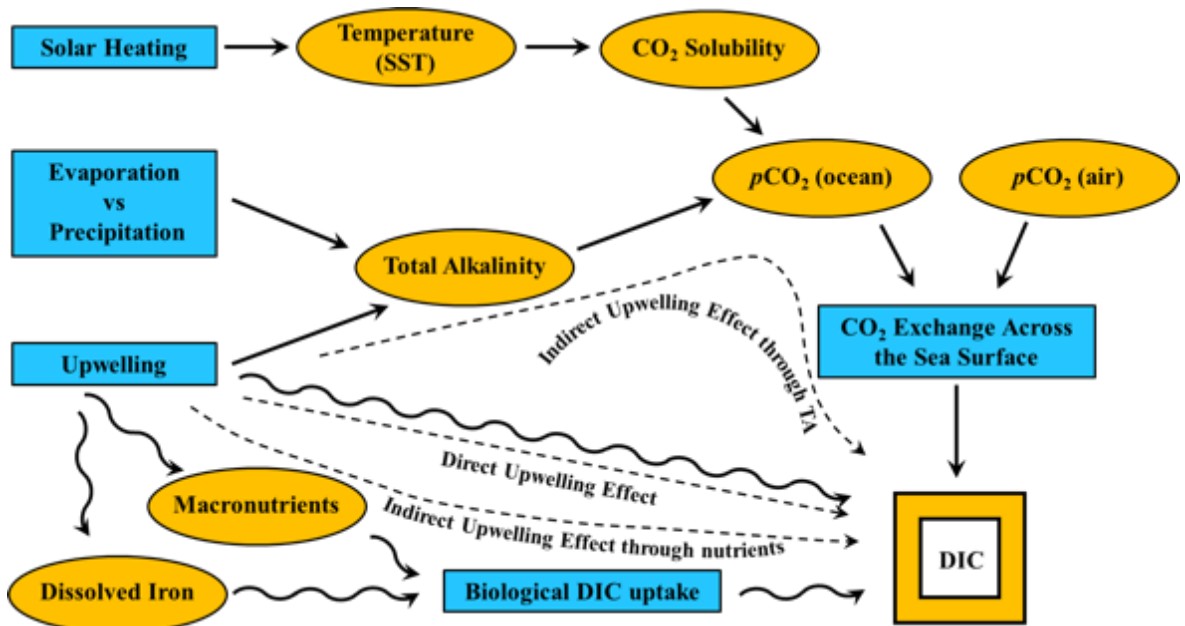

**Figure 2. Major controls on surface DIC.** Schematic showing the main processes exerting an influence over the concentration of DIC in the global surface ocean (producing variation with latitude). Blue shapes are processes and orange shapes are variables. Straight solid arrows represent equilibrium processes regulating nDIC in the long-term and wavy solid arrows represent disequilibrium processes regulating nDIC in the short-term. In the manuscript, we evaluate the upwelling effect on surface DIC in the Southern Ocean. Dashed arrows with text denote the three different ways that upwelling affects DIC: the direct effect through upwelled DIC; the indirect effect through upwelled nutrients which stimulates biological removal of DIC; and the indirect effect through upwelled TA in changing the equilibrium DIC with the atmosphere.

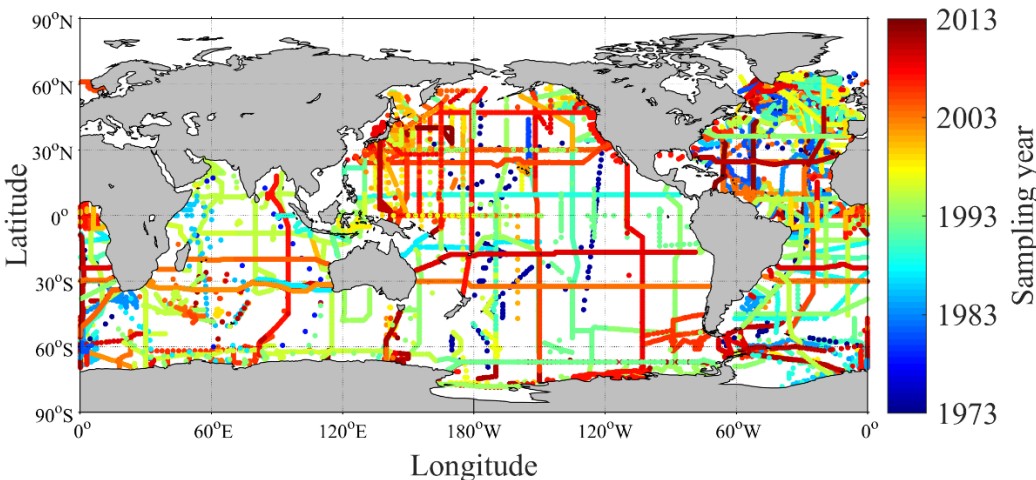

**Figure 3. Spatial and temporal distribution of GLODAPv2 sampling stations used for this study.**

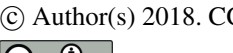



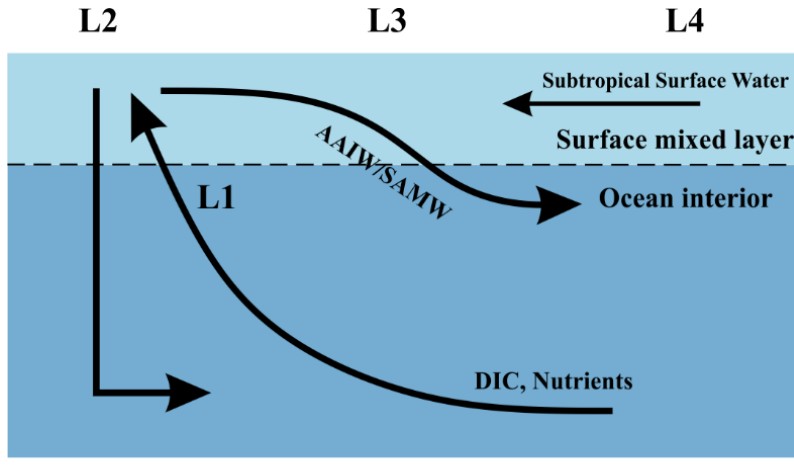

**Figure 4. A schematic illustrating locations of interest and assumed major flow paths in the Southern Ocean.** Black arrows represent the flow directions of water masses. The lower curved arrow denotes upwelling of deep water along isopycnal surfaces, and the upper curved arrow denotes subduction to form Subantarctic Mode Water (SAMW) and Antarctic Intermediate Water (AAIW). L1: upwelling water below the mixed layer, prior to any influence of surface processes; L2: sea surface within the core of the Southern Ocean upwelling south of 50° S (Morrison et al., 2015); L3: sea surface from 30° S to 50° S; L4: sea surface north of 30° S which experiences no direct effects from upwelling in the Southern Ocean.



**Figure 5. Vertical distributions of (a) nPhos, (b) Alk* and (c) nDIC along the Atlantic Ocean section.** The Indian and Pacific sections are not shown. The selected Atlantic section (A25W) is shown as the red line on the right-hand side of the inset. The neutral density isopycnal along which upwelling occurs is indicated by the white contour, with the black contours referring to the observed variable.



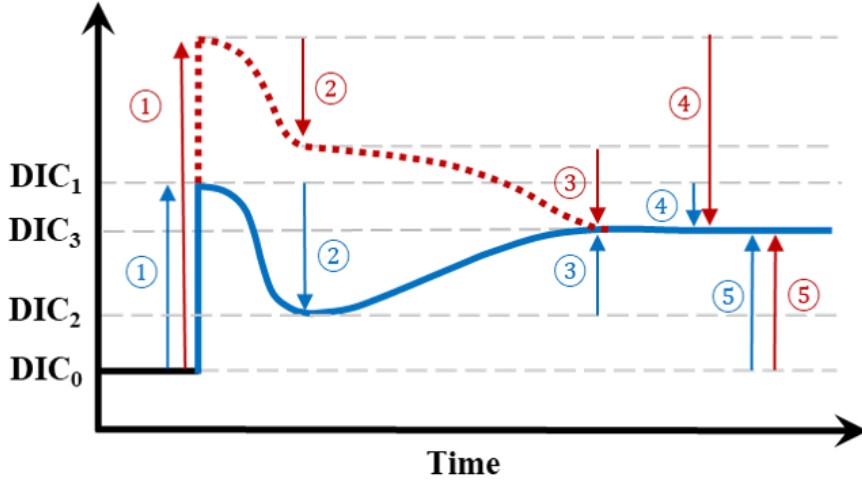

**Figure 6. A schematic illustrating the various effects of upwelling on surface DIC.** Numbers represent processes changing surface DIC, arrows point in the direction of change: ①: the *direct effect* of upwelling which elevates surface DIC from $DIC_0$ to $DIC_1$; ②: the DIC uptake by biology supported by upwelled nutrients, dropping DIC from $DIC_1$ to $DIC_2$. The processes of ① and ② make up the *short-term* effect of upwelling (i.e., difference between $DIC_2$ and $DIC_0$); ③: the change brought about by air-sea $CO_2$ gas exchange which continues towards the equilibrium with the atmosphere ($DIC_3$, whose level is determined by the amount of upwelled TA as well as by temperature); ④: the combination of both ② and ③ makes up the total *indirect effect* of upwelling (the difference between $DIC_3$ and $DIC_1$); ⑤: the *long-term* impact of upwelling on the level of surface DIC is the difference between $DIC_3$ and $DIC_0$. Blue and red indicate two scenarios with different amounts of upwelled DIC relative to upwelled TA, but the same amounts of upwelled TA. Blue is for upwelled water with a deficit in additional DIC relative to additional TA whereas red is for an excess in DIC relative to TA.





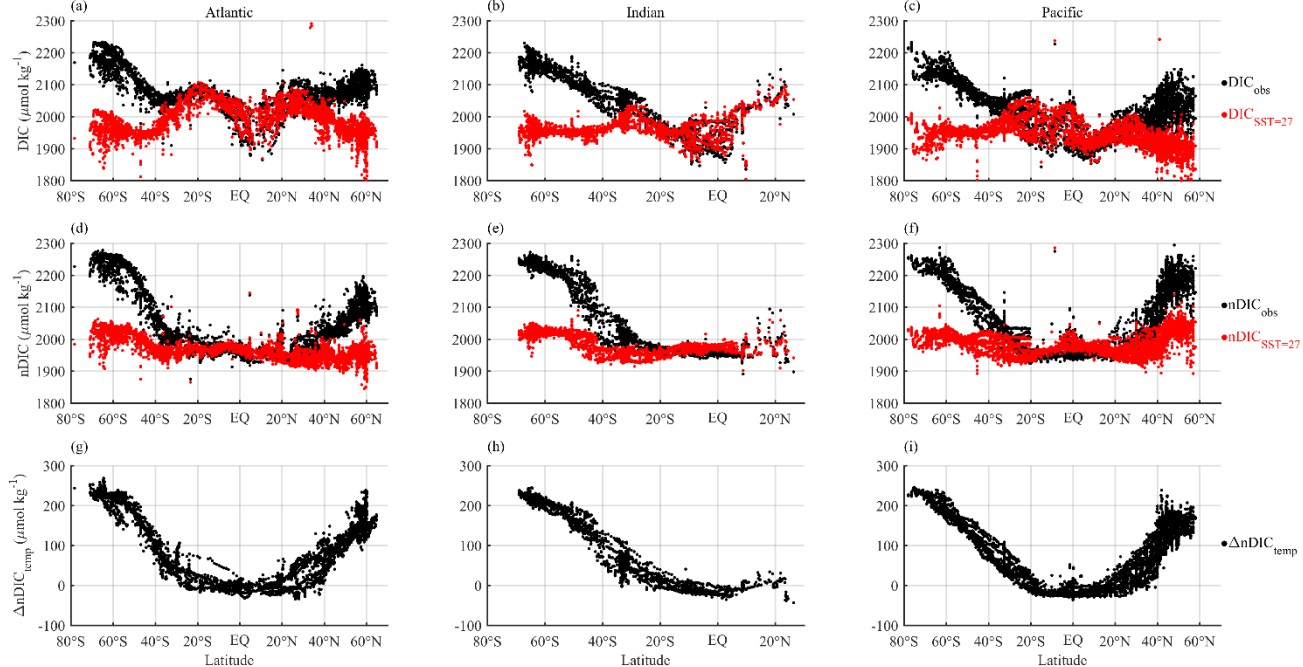

**Figure 7. Latitudinal distributions of calculated temperature effect on surface DIC.** Different columns show different basins (Atlantic, Indian and Pacific) and different rows show different calculated DIC variables. a, b and c show the observed surface DIC (black) and predicted DIC at SST of 27°C (red). d, e and f show the observed surface nDIC (black) and predicted nDIC at SST of 27°C (red). g, h and i show $\Delta nDIC_{temp}$, where $nDIC_{SST=27}$ is subtracted from $nDIC_{obs}$ to obtain the calculated temperature effect.



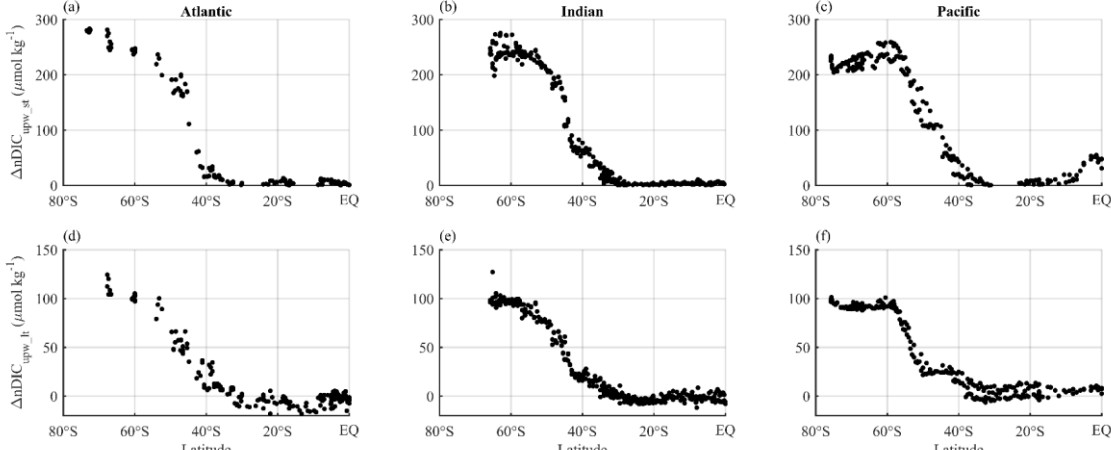

**Figure 8. Latitudinal distributions of calculated upwelling effects on surface nDIC.** Different columns show different sectors in ocean basins (Atlantic, Indian and Pacific) and different rows show different calculated effects on surface DIC. a, b and c show the short-term effect of upwelling ($\Delta nDIC_{upw\_st}$), which is driven by the direct supply of DIC from deep water and subsequent change by biology in the Southern Ocean. d, e and f show the long-term effect of upwelling ($\Delta nDIC_{upw\_lt}$), which is the difference between the observed nDIC value (determined mainly by the amount of upwelled TA, as well as by SST) and pre-upwelling nDIC value.



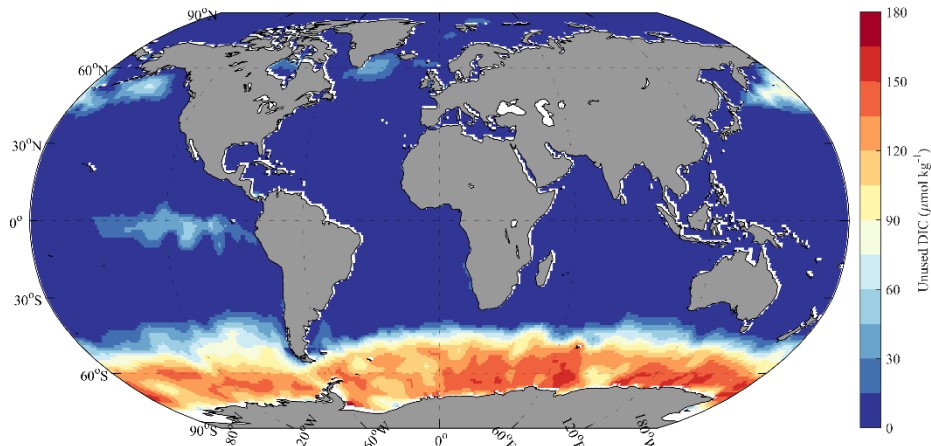

**Figure 9. Calculated potential impact of iron limitation on surface DIC.** Different colors correspond to different amounts of "unused DIC", calculated by Redfield ratio from observed residual nitrate.




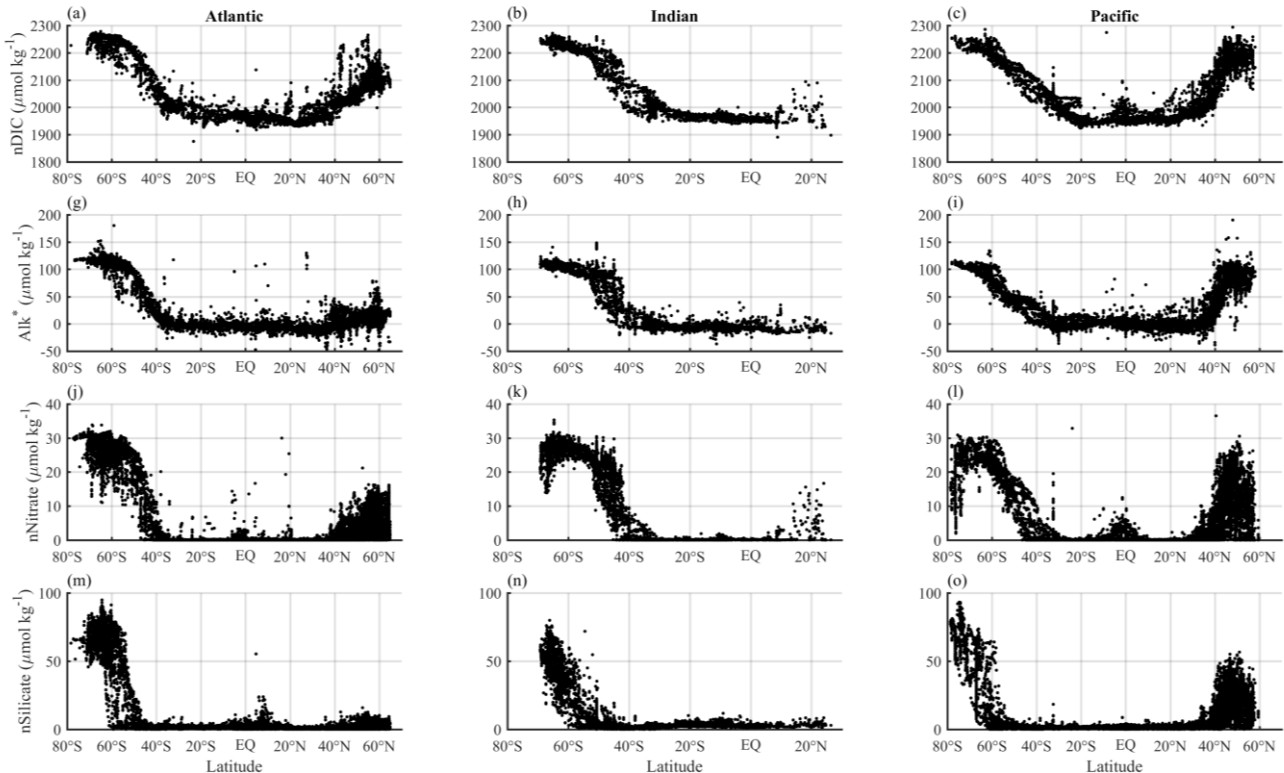

**Figure 10. Latitudinal distributions of sea surface nDIC, Alk* (*Fry et al.*, 2015), salinity-normalized nitrate and silicate in each ocean basin.**