# Peer review of "What drives the latitudinal gradient in open ocean surface dissolved inorganic carbon concentration?"

_Biogeosciences, 2018_

## Referee Comment (RC1) · Anonymous Referee #1 · 14 Sep 2018

General comments

Wu and colleagues present a detailed analysis based on the GLODAPv2 data-base of DIC distribution in oceanic surface waters, and unravel the processes responsible for latitudinal gradients.

Major comment

A major short-coming of the paper is the use of Equation 5 (Page 5) to normalize DIC. This procedure has been criticised in the past and shown to create artificial variance in DIC and TA distributions (Friis et al. 2003).

[Figure]

Specific comments

Page 1 L 28 : The formulation of the sentence gives the impression that the concept of DIC was defined by Zeebe and Wolf-Gladrow (2001) which is not the case.

Page 1 L 28 : You might want to define the * of CO2*

P 3 L5-13: I suggest to add calcification as a process controlling DIC, as global CaCO3 production from calcification is estimated at about 1.6 PgC/yr (Balch et al. 2007), nearly equivalent to the net oceanic sink of atmospheric CO2.

P 4 L 17: The formulation of the sentence gives the impression that the Mediterranean is "heavily influenced by river inputs". The Mediterranean Sea itself is an evaporative basin, with a salinity around 38 higher than the one of the North Atlantic. Only the Black Sea is "heavily influenced by river inputs".

P7L16: I think "organic matter burial" is more adequate than "organic matter sinking to the seafloor"

Page 8 Equation 10: Shouldn't you normalize to a constant salinity Alk_surf and Alk_supply prior to the calculation ? Rain or evaporation will change Alk_surf this will lead to a deviation from Alk_supply generating a value for Alk_CaCO3 even in absence of actual calcification.

Page 8 Equation 10: similarly Alk_surf should be corrected for the assimilation of NO3 and PO4 by primary production prior to the computation of Alk_CaCO3. Assimilation of inorganic nutrients by primary production leads to an increase of Alk_surf compared to Alk_supply (Brewer & Goldman 1976) that might obscure the actual signal from calcification.

Page 9: I do not understand how you accommodate equations 12 and 16 in your model. Equation 12 also accounts for NCP effect on DIC, so that if PO4 is exhausted this equation allows to compute for "unused DIC". If Fe is the limiting (micro)-nutrient for primary production then this should lead to "unused PO4" and allow to compute

"unused DIC" also from equation 12.

In addition, the authors preferred instead to use "unused nitrate" because "phosphorus has greater plasticity than nitrogen in plankton stoichiometry". Then this same reason should apply to compute the effect of NCP on DIC and prohibit the use of phosphate in equation 12, and nitrate should be preferred in equation 12.

For the sake of conceptual consistency in the approach it would be advisable that the effect of DIC assimilation by primary producers and NCP is computed in a consistent way independently of macro/micro nutrient limiting primary production.

References

Balch et al. (2007) Prediction of pelagic calcification rates using satellite measurements, Deep Sea Research Part II 54:478-495.

Brewer, P. G., and J. C. Goldman (1976), Alkalinity changes generated by phytoplankton growth, Limnol. Oceanogr., 21, 108-117.

Friis, K., A. KoÂĺrtzinger, and D. W. R. Wallace, The salinity normalization of marine inorganic carbon chemistry data, Geophys. Res. Lett., 30(2), 1085, doi:10.1029/2002GL015898, 2003.

---

## Author Comment (AC1) · 25 Sep 2018

We thank referee#1 for reviewing the paper. We address point by point the concerns of the referee.

**General comments:**

"A major short-coming of the paper is the use of Equation 5 (Page 5) to normalize DIC. This procedure has been criticised in the past and shown to create artificial variance in DIC and TA distributions (Friis et al. 2003)."

**Answer:**

We agree with the referee that the traditional method of salinity normalization can create artificial variance, as argued by Friis et al. (2003), because it attributes all salinity variation to evaporation and precipitation and ignores the influences of riverine input and upwelling from below the lysocline.

However, we excluded regions subjected to significant perturbations from river inputs (the ocean areas most likely to be affected, as suggested by Kang et al. (2013) and Fry et al. (2015)) as described in section 2.1.

In addition, no major rivers flow into the Southern Ocean. Therefore this is not a concern for our study.

The influence of upwelling from below the lysocline on DIC salinity normalization is also relatively negligible in our study. We compared values normalized using the Friis et al. (2003) method (taking account of upwelling) to our original values (normalised in the traditional way) and found a maximum difference of 3.5  $\mu$ mol kg-1, which is of the similar magnitude to the uncertainty in DIC measurement (4  $\mu$ mol kg-1) and much smaller than the phenomenon of interest (>100  $\mu$ mol kg-1).

The calculation following Friis et al. (2003) was implemented using an empirical relationship of the form nDIC =  $\frac{\text{DIC}^{\text{meas}} \cdot \text{DIC}^{\text{S=0}}}{\text{S}^{\text{meas}}} \cdot \text{S}^{\text{ref}} + \text{DIC}^{\text{S=0}}$ , where 'meas' denotes measured value and 'ref' denotes reference. This equation can then be converted into  $\text{nDIC} = \frac{\text{DIC}^{\text{meas}}}{\text{S}^{\text{meas}}} \cdot \text{S}^{\text{ref}} + \text{DIC}^{\text{S=0}} \cdot (1 - \frac{\text{S}^{\text{ref}}}{\text{S}^{\text{meas}}})$ , where the first term on the right side is the equation we used in the manuscript (Equation 5), and the second term on the right indicates the difference between the traditional salinity normalization and suggested salinity normalization by Friis et al. (2003). Take the Southern Ocean for instance where the largest upwelling in the world takes place. The magnitude of the difference term (i.e.,  $\text{DIC}^{\text{S=0}} \cdot (1 - \frac{\text{S}^{\text{ref}}}{\text{S}^{\text{meas}}})$ ) depends on  $\text{DIC}^{\text{S=0}}$  and  $\text{S}^{\text{meas}}$ .  $\text{DIC}^{\text{S=0}}$  is the region-specific term of S=0 due to upwelling of deep water which accumulates the remineralized inorganic carbon. Based on the GLODAPv2 database, the average concentration of DIC at depth greater than 500 m (the largest mixed layer depth in winter in the Southern Ocean, Dong et al., (2008)) in the Southern Ocean (>40^{\circ}\text{S}) is 2250 \,\mu\text{mol kg}^{-1}, and the average concentration of DIC at upwelling Ocean as defined in this study is 2130  $\mu\text{mol kg}^{-1}$ . As a consequence, upwelling from below the lysocline ( $\text{DIC}^{\text{S=0}}$ ) can create the largest DIC difference of 120  $\mu\text{mol kg}^{-1}$ .

1 (obviously DICS=0 in the modern surface ocean should be less than 120 µmol kg-1, otherwise the DIC vertical distribution would be uniform). The difference term  $\text{DIC}^{S=0} \cdot (1 - \frac{S^{\text{ref}}}{S^{\text{meas}}}))$  is always relatively small because the average measured salinity in the surface Southern Ocean is 34 and the reference salinity is 35 and so their ratio is close to 1.

**Specific comments:**

Page 1 L28: To avoid the wrong impression as pointed out by the referee, we removed the reference cited here since the concept of DIC has been defined very well already.

Page 1 L28: Done. Definition of  $[CO_2^*]$  added.

Page 3 L5-13: Done. We added "production and export of CaCO3" to be another process directly changing the surface distribution of DIC.

Page 4 L17: Corrected. Moved "the Mediterranean Sea" to the next line.

Page 7 L16: Changed.

Page 8 Equation 10: the calculation of Alk\* in this manuscript followed Fry et al. (2015), which is already salinitynormalized to a constant salinity of 35. The Alk\*surf in Equation 10 was calculated by Equation 11. We added a sentence for clarification.

Page 8 Equation 10: We agree with the referee that the assimilation of inorganic nutrients by primary production leads to an increase of  $Alk^*_{surf}$ . We ignored this process because the effect of primary production on TA is much smaller than that on DIC (TA:DIC = -17/106, Wolf-Gladrow et al., 2007). We now modify Equation 10 with a new term to account for the change in TA from photosynthesis or respiration. The latest calculated short-term effect of upwelling (revised Figure 8) is slightly different from the previous version.

Page 9: We thank the referee for pointing out the inconsistency in calculating NCP effect on DIC and calculating "unused DIC". We therefore decide to use phosphate as the only nutrient in Equation 12 and 16 in the next revised version.

**References**

- Dong, S., Sprintall, J., Gille, S. T., and Talley, L.: Southern Ocean mixed-layer depth from Argo float profiles, Journal of Geophysical Research: Oceans, 113, C06013, 10.1029/2006JC004051, 2008.
- Friis, K., Körtzinger, A., and Wallace, D. W.: The salinity normalization of marine inorganic carbon chemistry data, Geophysical Research Letters, 30, 1080, http://doi.org/10.1029/2002GL015898, 2003.
- Fry, C. H., Tyrrell, T., Hain, M. P., Bates, N. R., and Achterberg, E. P.: Analysis of global surface ocean alkalinity to determine controlling processes, Marine Chemistry, 174, 46-57, http://doi.org/10.1016/j.marchem.2015.05.003, 2015.
- Kang, Y., Pan, D., Bai, Y., He, X., Chen, X., Chen, C.-T.A., and Wang, D.: Areas of the global major river plumes. Acta Oceanologica Sinica, 32 (1), 79–88, http://dx.doi.org/10.1007/s13131-013-0269-5, 2013.
- Wolf-Gladrow, D. A., Zeebe, R. E., Klaas, C., Kortzinger, A., and Dickson, A. G.: Total alkalinity: The explicit conservative expression and its application to biogeochemical processes, Marine Chemistry, 106 (1-2), 287-300, http://doi.org/10.1016/j.marchem.2007.01.006, 2007.

---

## Referee Comment (RC2) · Anonymous Referee #2 · 9 Oct 2018

Wu et al. present a detailed analysis of factors driving the surface ocean concentration of dissolved inorganic carbon (DIC). Their study is based on the recently released GLODAP v2 dataset. In order to compare DIC in a global perspective they use salinity normalized DIC (NDIC). The major conclusion of their study is that sea surface temperature (SST) is the major driver of DIC variability, followed by changes in alkalinity and Southern Ocean upwelling. Major comments: Since the study is based on the normalization of DIC I'm wondering about the used normalization. It was shown that an easy division by salinity is problematic especially in a global perspective. The authors should validate their approach or at least discuss its problems. The authors use the GLODAP v2 dataset for the surface ocean. During their calculations they convert data

several times into pCO2. I'm wondering if the use of SOCAT as a pure surface ocean data set might be useful in order to check the calculations. What is with seasonality? Are the authors using seasonal average? I want to suggest merging the discussion and results sections. There are a lot of parameters discussed and described. Streamlining the sections and shorten it might help the readability. The whole manuscript is not easy to read. This is partly just to the fact that a lot of conversions are done and the reader had to keep track of it. There is not much that can be done to this. But the authors should carefully proof read their manuscript as some phenomena are discussed in several different positions of the manuscript. Again I want to advocate shortening it where possible in order to improve the readability. Some formulations sound odd and sometimes it's going back and forth especially in the introduction section. Specific comments (pp/ll): 01/24: cite the most actual GCB from 2017 or even 2018 01/29: I'm not sure, but is it worth to explain what CO2* is? 02/31 – 03/03: the authors discuss Takahashi (2014) work on page 2. On page 3 they say "since these studies the database was extended...". This doesn't make sense. 03/12: "other processes" sounds very broad. Can you specify? 03/14-16: repetition from before 03/18-20: Repetition from before 04/13: I prefer "water depth" over "seafloor depth" 04/25ff: Do the authors take spatial variability of atmospheric CO2 into account? 04/29: what is xCO2air? Please explain. 05/1-11: Somehow I got confused. It's a lot of steps for a quite easy process. But right now I also don't have a better solution. Just wanted to mention my first thought. 05/22ff: The formulation is odd. The authors state that they discuss the results in order of their hypotheses with exemptions. There are only three hypotheses so that sentence doesn't make sense to me. 06/10ff: The authors mention that the increased pCO2 has the potential to elevate values above atmospheric level. But it also can just lower the gradient if seawater is undersaturated. 07/03: "...Antarctic Circumpolar Current (ACC)..." 07/05: The term "L3" is not introduced. 07/25: One example of not thoroughly structured the document. The authors talk about phosphorus and Redfield. They don't give a number nor a reference. This comes with part of the discussion here later. Please merge. 07/33: Do you mean equation 10? 08/07: Together with Eq. 9

it reduces to nDICsurf = nDICsupply – NCP – 0.5xALK*CaCO3 08/09: RC should be RC:P 08/11: reference to Figure 5b is 5c 08/12ff: Presenting all the values in a table might be easier to read. 08/28: the effect has the potential to lower seawater pCO2 below atmospheric values. 09/32: Why are you not using the nitrate values from GLO-DAP? 11/01: Is evaporation only happening in the Atlantic? 11/15: Why is nDICtemp the gas exchange effect? Can you explain? 17/17ff: CDIAC is no longer maintained.

---

## Author Comment (AC2) · 20 Oct 2018

We thank referee#2 for reviewing the paper. We address point by point (answers in red) the concerns of the referee.

**General comments:**

Since the study is based on the normalization of DIC I'm wondering about the used normalization. It was shown that an easy division by salinity is problematic especially in a global perspective. The authors should validate their approach or at least discuss its problems.

We understand that the referee is concerned that the traditional salinity normalization would create artificial variance as argued by Friis et al. (2003), because the traditional calculation ignores the influences of riverine input and upwelling from below the lysocline.

This issue was addressed in our reply to Referee#1 (https://www.biogeosciences-discuss.net/bg-2018-376/bg-2018-376-AC1-supplement.pdf). We added one paragraph to Section 2.2 to explain why this issue is not quantitatively important in our study.

The authors use the GLODAP v2 dataset for the surface ocean. During their calculations they convert data several times into pCO2. I'm wondering if the use of SOCAT as a pure surface ocean data set might be useful in order to check the calculations.

We did not use SOCAT to check the calculation because it cannot provide a point by point comparison (SOCAT sampling points were at different times and places than GLODAPv2 sampling points).

Takahashi et al. (2014) found good agreement between calculated variables of the carbonate system (e.g., TA, DIC, $pCO_2$) compared to measured values. Values agreed within the measurement uncertainties assumed for the LDEO database. They found "the computed pCO2 values from the $TCO_2$ and TA data are in agreement with the measured values within ±6.8 µatm". We added one sentence in Section 2.4.1 to accommodate this issue.

What is with seasonality? Are the authors using seasonal average?

We did not discuss seasonality in the text because this is not the aim of this manuscript, and seasonal changes are of a much lower magnitude than the latitudinal gradient. For instance, the seasonal difference at the KERFIX site in the Southern Ocean was measured to be 25 µmol kg$^{-1}$ (Louanchi et al., 1999), which is much less than the overall latitudinal gradient we are looking to explain, of order 200 µmol kg$^{-1}$.

In addition, in general GLODAPv2 does not contain enough data to constrain seasonal cycles, particularly in the Southern Ocean where winter data is scarce.

I want to suggest merging the discussion and results sections. There are a lot of parameters discussed and described. Streamlining the sections and shorten it might help the readability. The whole manuscript is not easy to read. This

is partly just to the fact that a lot of conversions are done and the reader had to keep track of it. There is not much that can be done to this.

But the authors should carefully proof read their manuscript as some phenomena are discussed in several different positions of the manuscript. Again I want to advocate shortening it where possible in order to improve the readability. Some formulations sound odd and sometimes it's going back and forth especially in the introduction section.

We thank the referee for providing insights into how to improve the readability. In addition to the specific places where repetitions were found by the referee, we also shortened and modified the text (e.g. changes made in Section 1, Section 2.4, and Section 4.1) to avoid repetition and be clearer. However, we decided not to merge the results and discussion sections because it would lead to some very long sub-sections, which in our view would be off-putting to some readers and make the manuscript less rather than more readable.

**Specific comments (pp/ll):**

01/24: cite the most actual GCB from 2017 or even 2018.

Changed.

01/29: I'm not sure, but is it worth to explain what CO2* is?

We added an explanation.

02/31 – 03/03: the authors discuss Takahashi (2014) work on page 2. On page 3 they say "since these studies the database was extended: : :". This doesn't make sense.

We thank the referee for pointing out this issue. We have revised the sentence.

03/12: "other processes" sounds very broad. Can you specify?

We have modified the text to read "the above processes (1-4) which affect TA simultaneously".

03/14-16: repetition from before 03/18-20: Repetition from before.

03/14-16 actually summarizes the novelty, with which we expected to give the readers a straightforward view of what is new about this study. 03/18-20 was removed, along with 03/20-22.

04/13: I prefer "water depth" over "seafloor depth".

Changed.

04/25ff: Do the authors take spatial variability of atmospheric CO2 into account?

We used the globally averaged atmospheric $CO_2$. We now state this.

04/29: what is xCO2air? Please explain.

We explained in 04/27 when this term first appears. $xCO_{2,air}$ refers to the atmospheric mole fraction of $CO_2$. We

added an explanation to the superscript appeared in Equation 2.

05/1-11: Somehow I got confused. It's a lot of steps for a quite easy process. But right now I also don't have a better solution. Just wanted to mention my first thought.

The principle here is we assumed that sea surface $CO_2$ changes ($\Delta xCO_{2,sw}$) track atmospheric $CO_2$ changes ($\Delta xCO_{2,air}$). Then $\Delta xCO_{2,sw}$ was converted into $\Delta DIC$ using CO2SYS. We have made a couple of small changes that hopefully reduce confusion.

05/22ff: The formulation is odd. The authors state that they discuss the results in order of their hypotheses with exemptions. There are only three hypotheses so that sentence doesn't make sense to me.

We wanted to discuss the processes step by step. The second hypothesis can be assessed by salinity normalization so we did not have to repeat the analysis. We modified the text to be more accurate.

06/10ff: The authors mention that the increased pCO2 has the potential to elevate values above atmospheric level. But it also can just lower the gradient if seawater is undersaturated.

We agreed with the referee. This paragraph was just to give an explanation of how temperature changes (we used sea water warming as an example) alter the air-sea $CO_2$ gas exchange and therefore sea surface pCO$_2$ and DIC. We used the word "potentially" to indicate that this is just one aspect, which sounds most understandable to readers.

07/03: ": : :Antarctic Circumpolar Current (ACC): : :"

Changed.

07/05: The term "L3" is not introduced.

We defined L3 in the caption of Figure 4 and now cite that figure.

07/25: One example of not thoroughly structured the document. The authors talk about phosphorus and Redfield. They don't give a number nor a reference. This comes with part of the discussion here later. Please merge.

The number and reference were given after Equation 12 when it came into the calculation of nDIC$_{surf}$. We moved the number and reference for $R_{C:P}$ from 08/08 to 07/25.

07/33: Do you mean equation 10?

We thanked the referee for pointing out the mistake. Changed.

08/07: Together with Eq. 9 it reduces to nDICsurf = nDICsupply – NCP – 0.5xALK*CaCO3.

Changed.

08/09: RC should be RC:P.

Changed.

08/11: reference to Figure 5b is 5c.

Changed.

08/12ff: Presenting all the values in a table might be easier to read.

We have already got 4 tables and 10 figures, therefore we decided to keep these values in the text.

08/28: the effect has the potential to lower seawater pCO2 below atmospheric values.

Changed.

09/32: Why are you not using the nitrate values from GLODAP?

Because GLODAPv2 has sparse observations in the Southern Ocean. For this reason we used the global gridded product based on the WOA dataset. This much larger dataset is available for phosphate but not for DIC or TA. In response to a comment from referee#1, we decided in any case to use the phosphate rather than the nitrate data from WOA in order to be consistent with the calculation for NCP in Equation 9.

11/01: Is evaporation only happening in the Atlantic?

No, the main reason is the very large water vapour transport across Central America, due to the prevailing wind direction towards the west. To avoid giving a misleading impression, we removed the phrase "due to the intense evaporation in the subtropical Atlantic Ocean".

11/15: Why is nDICtemp the gas exchange effect? Can you explain?

$\Delta nDIC_{temp}$ is analogous to $\Delta DIC_{temp}$, which is defined in equation 8 (Section 2.4.1), It is a measure of the temperature-driven gas exchange effect because it is the difference between nDIC at in-situ temperature and nDIC at 27C. We have added a citation of equation 8 to clarify. As explained in section 2.4.1: "air-sea $CO_2$ gas exchange was assumed to proceed until $pCO_2$ was back to the same level as before resetting the temperature."

17/17ff: CDIAC is no longer maintained.

Changed.

**References**

Friis, K., Körtzinger, A., and Wallace, D. W.: The salinity normalization of marine inorganic carbon chemistry data, Geophysical Research Letters, 30, 1080, http://doi.org/10.1029/2002GL015898, 2003.

Louanchi, F., Ruiz-Pino, D. P., and Poisson, A.: Temporal variations of mixed-layer oceanic $CO_2$ at JGOFS-KERFIX time-series station: Physical versus biogeochemical processes, Journal of Marine Research, 57, 165-187, 10.1357/002224099765038607, 1999.

Takahashi, T., Sutherland, S. C., Chipman, D. W., Goddard, J. G., Ho, C., Newberger, T., Sweeney, C., and Munro, D. R.: Climatological distributions of pH, pCO2, total CO2, alkalinity, and CaCO3 saturation in the global surface ocean, and temporal changes at selected locations, Marine Chemistry, 164, 95-125, http://doi.org/10.1016/j.marchem.2014.06.004, 2014.

---

## Author Response (AR1)

Dear Professor Middelburg,

Thank you very much for your attention to this manuscript. As requested, we have thoroughly revised the manuscript in accordance with the reviewer comments (in fact we did this already as soon as the comments were posted).

We have considered the reviewer comments carefully and responded in full to them below. Our primary changes have been to: (1) clarify the question of the use of traditional salinity normalization on DIC, (2) shorten the manuscript to reduce the repetition and enhance the readability, and (3) remove the inconsistency in using nitrate and phosphate to calculate different effects on surface (n)DIC. We believe that these changes, together with other suggested edits, have substantially improved the manuscript.

Yours sincerely,
Yingxu Wu and on behalf of all co-authors

**Response to Referee#1**
**General comments:**
"A major short-coming of the paper is the use of Equation 5 (Page 5) to normalize DIC. This procedure has been criticised in the past and shown to create artificial variance in DIC and TA distributions (Friis et al. 2003)."

   **Response:** We agree with the referee that the traditional method of salinity normalization can create artificial variance, as argued by Friis et al. (2003), because it attributes all salinity variation to evaporation and precipitation and ignores the influences of riverine input and upwelling from below the lysocline.

However, we excluded regions subjected to significant perturbations from river inputs (the ocean areas most likely to be affected, as suggested by Kang et al. (2013) and Fry et al. (2015)) as described in section 2.1. In addition, no major rivers flow into the Southern Ocean. Therefore this is not a concern for our study.

The influence of upwelling from below the lysocline on DIC salinity normalization is also relatively negligible in our study. We compared values normalized using the Friis et al. (2003) method (taking account of upwelling) to our original values (normalized in the traditional way) and found a maximum difference of 3.5 µmol kg$^{-1}$, which is of the similar magnitude to the uncertainty in DIC measurement (4 µmol kg$^{-1}$) and much smaller than the phenomenon of interest (>100 µmol kg$^{-1}$).

The calculation following Friis et al. (2003) was implemented using an empirical relationship of the form $nDIC = \frac{DIC^{meas} - DIC^{S=0}}{S^{meas}} \cdot S^{ref} + DIC^{S=0}$, where 'meas' denotes measured value and 'ref' denotes reference. This equation can then be converted into $nDIC = \frac{DIC^{meas}}{S^{meas}} \cdot S^{ref} + DIC^{S=0} \cdot (1 - \frac{S^{ref}}{S^{meas}})$, where the first term on the right side is the equation we used in the manuscript (Equation 5), and the second term on the right side indicates the difference between the traditional salinity normalization and suggested salinity normalization by Friis et al. (2003). Take the Southern Ocean for instance where the

largest upwelling in the world takes place. The magnitude of the difference term (i.e., $DIC^{S=0} \cdot (1-\frac{S^{ref}}{S^{meas}})$) depends on $DIC^{S=0}$ and $S^{meas}$. $DIC^{S=0}$ is the region-specific term of S=0 due to upwelling of deep water which accumulates the remineralized inorganic carbon. Based on the GLODAPv2 database, the average concentration of DIC at depth greater than 500 m (the largest mixed layer depth in winter in the Southern Ocean, Dong et al., (2008)) in the Southern Ocean (>40°S) is 2250 µmol kg$^{-1}$, and the average concentration of DIC at surface layer in the Southern Ocean as defined in this study is 2130 µmol kg$^{-1}$. As a consequence, upwelling from below the lysocline ($DIC^{S=0}$) can create the largest DIC difference of 120 µmol kg$^{-1}$ (obviously $DIC^{S=0}$ in the modern surface ocean should be less than 120 µmol kg$^{-1}$, otherwise the DIC vertical distribution would be uniform). The difference term $DIC^{S=0} \cdot (1-\frac{S^{ref}}{S^{meas}})$) is always relatively small because the average measured salinity in the surface Southern Ocean is 34 and the reference salinity is 35 and so their ratio is close to 1.

We added one paragraph to Section 2.2 to explain why this issue is not quantitatively important in our study.

**Specific comments:**

**Page 1 L28:** The formulation of the sentence gives the impression that the concept of DIC was defined by Zeebe and Wolf-Gladrow (2001) which is not the case.

> **Response:** To avoid the wrong impression as pointed out by the referee, we removed the reference cited here since the concept of DIC has been defined very well already.

**Page 1 L28:** You might want to define the * of CO2*

> **Response:** Done. Definition of $[CO_2^*]$ added.

**Page 3 L5-13:** I suggest to add calcification as a process controlling DIC, as global CaCO3 production from calcification is estimated at about 1.6 PgC/yr (Balch et al. 2007), nearly equivalent to the net oceanic sink of atmospheric CO2.

> **Response:** Done. We added "production and export of $CaCO_3$" to be another process directly changing the surface distribution of DIC.

**Page 4 L17:** The formulation of the sentence gives the impression that the Mediterranean is "heavily influenced by river inputs". The Mediterranean Sea itself is an evaporative basin, with a salinity around 38 higher than the one of the North Atlantic. Only the Black Sea is "heavily influenced by river inputs".

> **Response:** Corrected. Moved "the Mediterranean Sea" to the next line.

**Page 7 L16:** I think "organic matter burial" is more adequate than "organic matter sinking to the seafloor"

> **Response:** Changed.

**Page 8 Equation 10:** Shouldn't you normalize to a constant salinity Alk_surf and Alk_supply prior to the calculation? Rain or evaporation will change Alk_surf this will lead to a deviation from Alk_supply generating a value for Alk_CaCO3 even in absence of actual calcification.

> **Response:** The calculation of Alk* in this manuscript followed Fry et al. (2015), which is already salinity-normalized to a constant salinity of 35. The $Alk^*_{surf}$ in Equation 10 was calculated by Equation 11. We added a sentence for clarification.

**Page 8 Equation 10:** similarly Alk_surf should be corrected for the assimilation of NO3 and PO4 by primary production prior to the computation of Alk_CaCO3. Assimilation of inorganic nutrients by primary production leads to an increase of Alk_surf compared to Alk_supply (Brewer & Goldman 1976) that might obscure the actual signal from calcification.

**Response:** We agree with the referee that the assimilation of inorganic nutrients by primary production leads to an increase of $Alk^*_{surf}$. We ignored this process previously because the effect of primary production on TA is much smaller than that on DIC (TA:DIC via primary production = -17/106, Wolf-Gladrow et al., 2007). We now modified Equation 10 with a new term to account for the change in TA from photosynthesis or respiration. The latest calculated short-term effect of upwelling (revised Figure 8) is slightly different from the previous version.

**Page 9:** I do not understand how you accommodate equations 12 and 16 in your model. Equation 12 also accounts for NCP effect on DIC, so that if PO4 is exhausted this equation allows to compute for "unused DIC". If Fe is the limiting (micro)-nutrient for primary production then this should lead to "unused PO4" and allow to compute "unused DIC" also from equation 12.

In addition, the authors preferred instead to use "unused nitrate" because "phosphorus has greater plasticity than nitrogen in plankton stoichiometry". Then this same reason should apply to compute the effect of NCP on DIC and prohibit the use of phosphate in equation 12, and nitrate should be preferred in equation 12.

For the sake of conceptual consistency in the approach it would be advisable that the effect of DIC assimilation by primary producers and NCP is computed in a consistent way independently of macro/micro nutrient limiting primary production.

**Response:** We thank the referee for pointing out the inconsistency in calculating NCP effect on DIC and calculating "unused DIC". We therefore decide to use phosphate as the only nutrient in Equation 12 and 16 in the revised version.

In addition, in general GLODAPv2 does not contain enough data to constrain seasonal cycles, particularly in the Southern Ocean where winter data is scarce.

**General comments:** I want to suggest merging the discussion and results sections. There are a lot of parameters discussed and described. Streamlining the sections and shorten it might help the readability. The whole manuscript is not easy to read. This is partly just to the fact that a lot of conversions are done and the reader had to keep track of it. There is not much that can be done to this.

But the authors should carefully proof read their manuscript as some phenomena are discussed in several different positions of the manuscript. Again I want to advocate shortening it where possible in order to improve the readability. Some formulations sound odd and sometimes it's going back and forth especially in the introduction section.

**Response:** We thank the referee for providing insights into how to improve the readability. In addition to the specific places where repetitions were found by the referee, we also shortened and modified the text (e.g. changes made in Section 1, Section 2.4, and Section 4.1) to avoid repetition and be clearer. However, we decided not to merge the results and discussion sections because it would lead to some very long sub-sections, which in our view would be off-putting to some readers and make the manuscript less rather than more readable.

**Specific comments (pp/ll):**

**01/24:** cite the most actual GCB from 2017 or even 2018.

**Response:** Changed.

**01/29:** I'm not sure, but is it worth to explain what CO2* is?

**Response:** We added an explanation.

**02/31 – 03/03:** the authors discuss Takahashi (2014) work on page 2. On page 3 they say "since these studies the database was extended: : :". This doesn't make sense.

**Response:** We thank the referee for pointing out this issue. We have revised the sentence.

**03/12:** "other processes" sounds very broad. Can you specify?

**Response:** We have modified the text to read "the above processes (1-4) which affect TA simultaneously".

**03/14-16:** repetition from before 03/18-20: Repetition from before.

**Response:** 03/14-16 actually summarizes the novelty, which we think is important to retain in order to give the readers a straightforward view of what is new about this study. 03/18-20 was removed, along with 03/20-22.

**04/13:** I prefer "water depth" over "seafloor depth".

**Response:** Changed.

**04/25ff:** Do the authors take spatial variability of atmospheric CO2 into account?

**Response:** We used the globally averaged atmospheric $CO_2$. We now state this.

**04/29:** what is xCO2air? Please explain.

**Response:** We explained in 04/27 where this term first appears. $xCO_{2,air}$ refers to the atmospheric mole fraction of $CO_2$. We added an explanation after Equation 2.

**05/1-11:** Somehow I got confused. It's a lot of steps for a quite easy process. But right now I also don't have a better solution. Just wanted to mention my first thought.

**Response:** The principle here is we assumed that sea surface $CO_2$ changes ($\Delta xCO_{2,sw}$) track atmospheric $CO_2$ changes ($\Delta xCO_{2,air}$). Then $\Delta xCO_{2,sw}$ was converted into $\Delta DIC$ using CO2SYS. We have made a couple of small changes that hopefully reduce confusion.

**05/22ff:** The formulation is odd. The authors state that they discuss the results in order of their hypotheses with exemptions. There are only three hypotheses so that sentence doesn't make sense to me.

**Response:** We wanted to discuss the processes step by step. The second hypothesis can be assessed by salinity normalization so we did not have to repeat the analysis. We modified the text to be more clear.

**06/10ff:** The authors mention that the increased pCO2 has the potential to elevate values above atmospheric level. But it also can just lower the gradient if seawater is undersaturated.

**Response:** We agreed with the referee. This paragraph was just to give an explanation of how temperature changes (we used sea water warming as an example) alter the air-sea $CO_2$ gas exchange and therefore sea surface $pCO_2$ and DIC. We used the word "potentially" to indicate that this is just one possibility, which sounds most understandable to readers.

**07/03:** ": : :Antarctic Circumpolar Current (ACC): : :"

**Response:** Changed.

**07/05:** The term "L3" is not introduced.

**Response:** We defined L3 in the caption of Figure 4 and now cite that figure.

**07/25:** One example of not thoroughly structured the document. The authors talk about phosphorus and Redfield. They don't give a number nor a reference. This comes with part of the discussion here later. Please merge.

**Response:** The number and reference were given after Equation 12 when it came into the calculation of $nDIC_{surf}$. We moved the number and reference for $R_{C:P}$ from 08/08 to 07/25.

**07/33:** Do you mean equation 10?

**Response:** We thanked the referee for pointing out the mistake. Changed.

**08/07:** Together with Eq. 9 it reduces to nDICsurf = nDICsupply – NCP – 0.5xALK*CaCO3.

**Response:** Changed.

**08/09:** RC should be RC:P.

**Response:** Changed.

**08/11:** reference to Figure 5b is 5c.

**Response:** Changed.

**08/12ff:** Presenting all the values in a table might be easier to read.

**Response:** We have already got 4 tables and 10 figures, therefore we decided to keep these values in the text.

**08/28:** the effect has the potential to lower seawater pCO2 below atmospheric values.

**Response:** Changed.

**09/32:** Why are you not using the nitrate values from GLODAP?

**Response:** Because GLODAPv2 has sparse observations in the Southern Ocean. For this reason we used the global gridded product based on the WOA dataset. This much larger dataset is available for phosphate but not for DIC or TA. In response to a comment from referee#1, we decided in any case to use the phosphate rather than the nitrate data from WOA in order to be consistent with the calculation for NCP in Equation 9.

**11/01:** Is evaporation only happening in the Atlantic?

**Response:** No, the main reason is the very large water vapour transport across Central America, due to the prevailing wind direction towards the west. To avoid giving a misleading impression, we removed the phrase "due to the intense evaporation in the subtropical Atlantic Ocean".

**11/15:** Why is nDICtemp the gas exchange effect? Can you explain?

**Response:** $\Delta nDIC_{temp}$ is analogous to $\Delta DIC_{temp}$, which is defined in equation 8 (Section 2.4.1), It is a measure of the temperature-driven gas exchange effect because it is the difference between nDIC at in-situ temperature and nDIC at 27C. We have added a citation of equation 8 to clarify. As explained in section 2.4.1: "air-sea $CO_2$ gas exchange was assumed to proceed until $pCO_2$ was back to the same level as before resetting the temperature."

**17/17ff:** CDIAC is no longer maintained.

**Response:** Changed.

**References**

[revised manuscript text omitted]

---

## Referee Report (RR1)

Review of "What drives the latitudinal gradient in open ocean surface dissolved inorganic carbon concentration?" by Yingxu Wu et al.

The authors present an analysis of driving mechanisms behind the north-south gradient of surface ocean DIC. The study is based on DIC and alkalinity data from the database GLODAP v2.

Major issues:
The authors calculate pCO2 from DIC and TA data, which results in a non-equilibrium pCO2. This is then converted to a standard temperature and calculated back to TA and DIC assuming that pCO2 is back to its initial value. As a major part of the pCO2 disequilibrium in the ocean is associated to temperature changes wouldn't it be more useful to assume equilibrium with the atmosphere (at in-situ temperature as well as at standard temperature) when calculating the temperature effect on DIC?
The two major weaknesses raised during the first review iteration are still persisting, the normalization of DIC without a freshwater component and the disregard of seasonality.
I am not convinced that excluding areas strongly influenced by riverine input is enough to solve the problem with the DIC normalization. Fitting the alkalinity against the salinity gives also in the open ocean a non-zero intercept that can be vary from region to region. With the GLODAP v2 dataset there is an excellent database to calculate this intercept and it's regional variations. It has at least to be discussed what influence this would have on the latitudinal gradients.
Regarding the seasonality I do understand that there is not enough data in the Southern Ocean to resolve and discuss seasonalities. But nevertheless these influences need to be discussed. The salinity normalization of phosphate, for example, can produce relatively high residual phosphate and thus 'unused DIC' during summer in regions with low salinities although these might experience phosphate limitation during summer. Also I am missing a notice that using phosphate concentrations as tracer for the influence of primary production is a simplification. C:P ratios are highly variable, not only, as mentioned, regionally, but also seasonally. Under phosphate limitation phytoplankton is still able to fix carbon, and release it as DOC.
As a last major point, the manuscript is partly very difficult to read. The discussion of effects on DIC and nDIC is sometimes confusing and the reader might get lost which of both is discussed and why.

Minor issues:
p. 10 l. 21: Here $DIC_{obs}$ and $nDIC_{obs}$ are discussed, right?

p. 11 l. 12: Rephrase this sentence. It reads as if you have data at 70°S in the northern Atlantic.

p. 12 l. 13: This part needs some changes. Most of the DIC variability is in fact not caused by changes in alkalinity but by dilution and evaporation and therefore is

not existent anymore when discussing the nDIC. Alkalinity and DIC are just influenced in a similar manner.

p. 12 l. 25: TA and salinity are only in the open ocean highly correlated

p. 13 l. 9: nutrients, **nDIC** and nTA?

p. 13 l. 10: delete 'then'

p. 13 l. 30: change of nTA, or change of TA relative to the DIC changes

p. 14 l. 22: this should also be a function of the chemical composition: how low is the iron concentration in the upwelled water in comparison to the other nutrients.

p. 14 l. 23: delete '.'

Figure 1: increase the resolution of this picture. The description at the colour axis only says DIC

Figure 2: Here a direct influence from evaporation & precipitation to DIC is missing. Also, in the description of the figure it is written about both DIC and nDIC. For which of these two is the diagram?

Figure 5: The resolution of the picture should be improved. The map should be deleted from this picture. Show the location of the transects in Figure 3 instead. I think all figures should follow a common design when it comes to setup and fonts. Please change the design of this figure match the other figures. I don't understand the use of the black contour lines. Either, choose them in a way that the cover the entire data range or delete them.

Figure 8:  Are these the same transects as in figure 5?

Figure 9: Change the rotation of the longitudinal label to 0°.

---

## Author Response (AR2)

**Author's Response**

We thank referee#3 for reviewing the paper and for providing helpful comments which have improved the paper. We address point by point (answers in red) the concerns of the referee.

**Major issues:**

The authors calculate pCO2 from DIC and TA data, which results in a nonequilibrium pCO2. This is then converted to a standard temperature and calculated back to TA and DIC assuming that pCO2 is back to its initial value. As a major part of the pCO2 disequilibrium in the ocean is associated to temperature changes wouldn't it be more useful to assume equilibrium

10  with the atmosphere (at in-situ temperature as well as at standard temperature) when calculating the temperature effect on DIC?

We understand that the major part of the $pCO_2$ disequilibrium in the ocean is associated with temperature changes, with disequilibria able to persist for many months (e.g. Körtzinger et al. 2008) because of the long air-sea gas exchange

15  equilibrium timescale of $CO_2$. Previous studies such as Takahashi et al. (2014) revealed that in the real ocean, exact $CO_2$ equilibrium between the sea surface and the atmosphere above is hardly ever reached. Thus, the assumption of air-sea $CO_2$ equilibrium is a theoretical concept but is not often a practical reality. In fact we previously calculated temperature effects assuming air-sea equilibrium and obtained very similar results (the average absolute difference between temperature effects is about 20 µmol kg$^{-1}$). However, for the reasons just stated, we prefer to calculate the effects from the starting point of the

20  observed disequilibria.

The two major weaknesses raised during the first review iteration are still persisting, the normalization of DIC without a freshwater component and the disregard of seasonality.

I am not convinced that excluding areas strongly influenced by riverine input is enough to solve the problem with the DIC

25  normalization. Fitting the alkalinity against the salinity gives also in the open ocean a non-zero intercept that can be vary from region to region. With the GLODAP v2 dataset there is an excellent database to calculate this intercept and its regional variations. It has at least to be discussed what influence this would have on the latitudinal gradients.

We agree with the referee that more details about salinity normalization should be stated. As a supplement to our response to

30  the first referee (https://www.biogeosciences-discuss.net/bg-2018-376/bg-2018-376-AC1-supplement.pdf), which mainly focused on neglecting the influence of upwelling on DIC salinity normalization, we quantify here the impact of the 'non-zero intercept' issue raised by the referee.

Figure R1 shows the relationship between surface DIC and salinity. We selected data between 30°S and 30°N (i.e., the oligotrophic surface oceans) in order to avoid the perturbations of upwelling (this has been discussed in the response letter to

Referee#1, see link above) and biological activities on surface DIC. By fitting surface DIC against salinity, a non-zero intercept was found for each of the ocean basins, ranging from -187 µmol kg$^{-1}$ (Pacific Ocean) to 262 µmol kg$^{-1}$ (Atlantic). By substituting the values into Equation $nDIC = \frac{DIC^{meas} - DIC^{S=0}}{S^{meas}} \cdot S^{ref} + DIC^{S=0}$ (Friis et al., 2003), which can be further converted into $nDIC = \frac{DIC^{meas}}{S^{meas}} \cdot S^{ref} + DIC^{S=0} \cdot (1 - \frac{S^{ref}}{S^{meas}})$, we found that the discrepancy (i.e., the term $DIC^{S=0} \cdot (1 - \frac{S^{ref}}{S^{meas}})$) between the Friis

5  et al. (2013) method and our calculation (Equation 5 in the manuscript) only accounts for a difference ranging from -5 µmol kg$^{-1}$ to 7 µmol kg$^{-1}$, which is of similar magnitude to the uncertainty in DIC measurement and also much smaller than the phenomenon of interest (about 200 µmol kg$^{-1}$).

We have added some texts in Section 2.2 to discuss this influence.

[Figure]

10  Figure R1. Relationship between sea surface DIC and salinity in the Atlantic, Indian, and Pacific Oceans. The black dashed lines are best-fit linear regression lines.

Regarding the seasonality I do understand that there is not enough data in the Southern Ocean to resolve and discuss seasonalities. But nevertheless these influences need to be discussed.

We agree that seasonality in surface DIC and nDIC at high latitudes must result in seasonal variation of the nDIC latitudinal gradient. We have added a new section 4.1.4 to discuss the issue raised by the referee.

The salinity normalization of phosphate, for example, can produce relatively high residual phosphate and thus 'unused DIC'

20  during summer in regions with low salinities although these might experience phosphate limitation during summer.

We calculated 'unused DIC' based on the phosphate concentration without salinity normalization.

Also I am missing a notice that using phosphate concentrations as tracer for the influence of primary production is a simplification. C:P ratios are highly variable, not only, as mentioned, regionally, but also seasonally. Under phosphate limitation phytoplankton is still able to fix carbon, and release it as DOC.

We thank the referee for pointing out the seasonal variation of marine C:N:P stoichiometry (e.g., Frigstad et al., 2011). However, since the seasonal variation of C:P is much smaller than the latitudinal variation of C:P, which we care about most in this context, and since there have been no studies investigating the seasonal variation of C:P over the global ocean, we decided to only consider its latitudinal variation in this study. We have modified the text in Section 2.4.2 to acknowledge the

10    seasonal variation of C:P.

As a last major point, the manuscript is partly very difficult to read. The discussion of effects on DIC and nDIC is sometimes confusing and the reader might get lost which of both is discussed and why.

15    An English-speaking co-author has gone through the manuscript again in an attempt to improve readability. We have modified the text (e.g., revisions to the Introduction, Section 4.1 and Section 4.2) to try and make it as simple and understandable as possible given the complexity of the topic. We have added a short paragraph at the start of Section 4.1 to make it clear that the calculated effects are all based on nDIC.

20    **Minor issues:**

p. 10 l. 21: Here $DIC_{obs}$ and $nDIC_{obs}$ are discussed, right?

Yes, Section 3.1 described the distributions of $DIC_{obs}$ and $nDIC_{obs}$. The heading of this section has been modified to "Spatial distributions of observed DIC and nDIC".

p. 11 l. 12: Rephrase this sentence. It reads as if you have data at 70˚S in the northern Atlantic.

Changed.

30    p. 12 l. 13: This part needs some changes. Most of the DIC variability is in fact not caused by changes in alkalinity but by dilution and evaporation and therefore is not existent anymore when discussing the nDIC. Alkalinity and DIC are just influenced in a similar manner.

We agree that both mechanisms (direct and indirect effects of evaporation and precipitation) must operate but argue that the latter must win out in the end. One of the novel contributions of our paper, we believe, is that the long-term (> 1 or 2 years) consequences of upwelling for DIC concentrations are dictated by the upwelling-induced changes to TA, rather than by the immediate upwelling-induced changes to nutrient or DIC concentrations (Section 4.2). By the same logic (Figure 6), changes to TA from other causes, including from evaporation or precipitation, should also eventually, because of gas exchange, determine changes to equilibrium DIC. Even in an imaginary scenario where evaporation and precipitation had no direct effect on the DIC concentration, but only on the TA concentration, then the processes would still end up altering the DIC concentration through the causal sequence: evaporation or precipitation → TA change → pCO2 change → generation of air-sea CO2 disequilibrium → air-sea exchange of CO2 → DIC change until air-sea equilibrium is re-established. Figure 6 has been modified to acknowledge that evaporation / precipitation also has a direct impact on DIC.

p. 12 l. 25: TA and salinity are only in the open ocean highly correlated.

The phrase "surface ocean" has been changed to "surface open ocean" (as described in the methods section, our study is restricted to open ocean locations where the seafloor depth is greater than 200m).

p. 13 l. 9: nutrients, nDIC and nTA?

Changed.

p. 13 l. 10: delete 'then'

Changed.

p. 13 l. 30: change of nTA, or change of TA relative to the DIC changes.

Arrow ⑤ in Figure 6 is the difference between the original, pre-upwelling DIC and the final DIC after the long-term alkalinity effect has run to completion. The size of this difference is dictated by the TA (not nTA) change brought about by upwelling. Its magnitude is independent of the immediate DIC change brought about by upwelling (arrow ① in Figure 6). The text is correct as it stands and has not been changed.

p. 14 l. 22: this should also be a function of the chemical composition: how low is the iron concentration in the upwelled water in comparison to the other nutrients.

We relate the long-term effect of upwelling only to the surface TA and temperature, which together determine the surface DIC if the system is to be in equilibrium with the atmosphere. The iron concentration in the upwelled water was not considered in the manuscript because we used the changes in surface phosphate to calculate the effect of biological removal (i.e., if there was a complete absence of iron in the upwelled water, then biological uptake would not take place and phosphate concentrations would not alter from the original value).

p. 14 l. 23: delete '.'

Changed.

Figure 1: increase the resolution of this picture. The description at the colour axis only says DIC.

Figure 1 will have the required resolution when published. In Word and pdf files the figures are automatically compressed, making the resolution of the figures seem not good enough. When the figures are taken from separate uploaded files, as they are for publication, then the issue does not occur. The colour axis description has been changed to "DIC or nDIC".

Figure 2: Here a direct influence from evaporation & precipitation to DIC is missing. Also, in the description of the figure it is written about both DIC and nDIC. For which of these two is the diagram?

We added "also influence DIC" to the arrow from E. vs P. to TA. The diagram illustrates processes affecting DIC and we have changed the text to say this.

Figure 5: The resolution of the picture should be improved. The map should be deleted from this picture. Show the location of the transects in Figure 3 instead. I think all figures should follow a common design when it comes to setup and fonts. Please change the design of this figure match the other figures. I don't understand the use of the black contour lines. Either, choose them in a way that the cover the entire data range or delete them.

Figure setup and fonts were changed. The black contours highlighted the values of the selected variable along the upwelling isopycnal but we have now removed the black contours to keep the figure simple and avoid confusing readers. However, we decided to leave the map in Figure 5 for two reasons: (1) adding the transects to Figure 3 will cover up the information which shows the sampling year; (2) we selected the three transects only for the calculation of upwelling effects, therefore there is no need to show the transects before Section 2.4.2.

Figure 8: Are these the same transects as in figure 5?

Yes. All the associated calculations of upwelling effects are based on the three transects. We added one sentence in Section 3.3 as well as in the caption of Fig. 8 to state this.

Figure 9: Change the rotation of the longitudinal label to 0˚.

Changed.

[revised manuscript text omitted]

---

## Author Response (AR3)

**We thank referee#2 for reviewing the paper. We address point by point (answers in red) the concerns of the referee.**

Review of manuscript submitted by Wu et al. "What drives the latitudinal gradient in open ocean surface dissolved inorganic carbon concentration?"

Wu et al. present a detailed analysis of factors driving the surface ocean concentration of dissolved inorganic carbon (DIC). Their study is based on the recently released GLODAP v2 dataset. In order to compare DIC in a global perspective they use salinity normalized DIC (nDIC). The major conclusion of their study is that sea surface temperature (SST) is the major driver of DIC variability, followed by changes in alkalinity and Southern Ocean upwelling. Their conclusion points out that the upwelling effect in is ignored in global models but can change the DIC fields substantially.

This is my second review and I want to acknowledge that the manuscript improved and the authors took the comments of all reviewers serious. I don't add specific comments in this reviews as I don't think that this will improve the text as it might be even to the taste of the reader/author how to formulate certain sentences.

My major issues with the manuscript are:

(1) The normalization of DIC by salinity. I understand the authors motivation to do that, but I'm still wondering if the normalization of a parameter that is influenced by non-salinity related factors weakens their findings too much. What is the introduced error of this simple normalization?

The straightforward error/uncertainty and the artificial impact introduced by salinity normalization have been acknowledged in the manuscript. They are both very minor (~4 to 7 µmol DIC kg$^{-1}$, Table 2 and Section 2.2) compared to the phenomenon of interest.

**For the calculation of the SST-driven effect**, we first calculated the SST-driven effect on DIC, and then normalized it to calculate the effect on nDIC. In the figure below (Fig. R1, which we have taken into consideration in the early version of this manuscript), we compare the difference between $\Delta DIC_{temp}$ and $\Delta nDIC_{temp}$. Overall the difference is very close to zero except in the polar regions and even there it is less than 10 µmol kg$^{-1}$. 10 µmol kg$^{-1}$ is fairly small compared to the magnitude of the SST-driven effect at high latitudes (more than 220 µmol kg$^{-1}$). In order to avoid adding complexity, we did not explain this in the text.

[Figure]

Figure R1. The difference (= $\Delta DIC_{temp}$ - $\Delta nDIC_{temp}$) between $\Delta DIC_{temp}$ and $\Delta nDIC_{temp}$ in each ocean basin.

**For the calculation of the upwelled DIC-driven effect**, the equations were based on the salinity-normalized concept in order to correct for the influence of the vertical salinity variation. The only possible artificial error introduced was from the term '35/S'. The expected equation to calculate NCP in Equation 10 should be:

$Phos_{surf} = Phos_{supply}$ - $NCP/R_{C:P}$

Instead of

$$nPhos_{surf} = nPhos_{supply} - NCP/R_{C:P}$$

which can be rewritten as:

$$Phos_{surf} \times \frac{35}{S_{surf}} = Phos_{supply} \times \frac{35}{S_{supply}} - NCP/R_{C:P}$$

Since the salinity in the Southern Ocean (both $S_{surf}$ and $S_{supply}$) is always very close to 35, salinity normalization always results in small changes there. Repeating the calculations of NCP both with and without salinity normalisation and propagating the results into the calculations of $\Delta DIC_{upw}$ show that the effect is less than 4 µmol kg$^{-1}$.

**For the calculation of the upwelled TA-driven effect**, we also first calculated it based on the effect on DIC and then followed by the salinity normalization, the same as the procedure for calculating the sensitivity of the SST-driven effect. Again, because Southern Ocean salinity ≈ 35, the associated uncertainties are small (maximum of 5 µmol kg$^{-1}$ in the Antarctic area).

We have added some text to the manuscript to acknowledge these uncertainties.

(2) Readability: In parts of the manuscript the authors jump between DIC, nDIC, DICobs. Is DICFe the same as unused DIC? I'm not sure if the reader has always the right DIC in mind.

$\Delta DIC_{Fe}$ is indeed the same as 'unused DIC'. For simplicity, we have modified the text by removing the term 'unused DIC' – we now just use $\Delta DIC_{Fe}$. Changes were also made to remind the readers of the right DIC definitions, and we explain each different variable after the equation in which it first appears.

(3) The calculation of the effects is based on conversions to and from surface ocean pCO2 and atmospheric pCO2. Neither seasonality (air and water pCO2) nor latitudinal gradients (air pCO2) are considered. The latitudinal gradient in the atmospheric xCO2 can be easily more than 10 ppm. Thus I expect the introduced error in DpCO2 calculations and further DIC calculations to be higher than estimated.

The way we normalized DIC to a reference year of 2005 was primarily based on Equation 2 (see manuscript), where $\Delta xCO_{2,air}$ was calculated based on the Mauna Loa $CO_2$ record. In fact, the latitudinal gradient of air $CO_2$ is not a concern. As shown in Figure R2, there is no significant latitudinal gradient of air $CO_2$. Annual average air $CO_2$ is the same at all latitudes and is increasing year on year at the same rate at all latitudes.

The seasonality of air $xCO_2$ can potentially be a concern but the seasonality of water $pCO_2$ is not because it was calculated with the in-situ data. Figure R2 shows that the largest seasonal amplitude of atmospheric $xCO_2$ can reach up to 8 ppm at Mauna Loa (we excluded the Arctic region in this study, corresponding to Barrow in Alaska). This is fairly small compared to the long-term increase of atmospheric $xCO_2$ of ~90 ppm over 40 years during the observational period of GLODAPv2. If we assume a water parcel with the global average sea surface conditions for the **year 1980** (salinity of 34.6, temperature of 15 °C, TA of 2300 µmol kg$^{-1}$, and atmospheric $xCO_2$ of 340 ppm), by normalizing the carbonate system of this water parcel to the **reference year of 2005** (atmospheric $xCO_2$ around 380 ppm), we have to correct for a $\Delta xCO_{2,air}$ of 40 ppm over 25 yrs. So it can be seen that we correct for the larger difference (40 ppm) but not the smaller (8 ppm). Ignoring seasonality in atmospheric $xCO_2$ results in an introduced error in the DIC calculations of around ±2 µmol kg$^{-1}$. This magnitude is less than the uncertainty of DIC in GLODAPv2 dataset. We have now modified the text to acknowledge this introduced error.

[Figure]

Figure R2. Trends in atmospheric $CO_2$. The figure shows daily averaged $CO_2$ from four GMD Baseline observatories; Barrow, Alaska (in blue), Mauna Loa, Hawaii (in red), American Samoa (in green), and South Pole, Antarctica (in yellow). The thick black lines represent the average of the smoothed seasonal curves and the smoothed, de-seasonalized curves for each of the records. These lines are a very good estimate of the global average levels of $CO_2$. Figure from Earth System Research Laboratory (https://www.esrl.noaa.gov/gmd/ccgg/trends/gl_trend.html).

I suggest to check the paper for possible sections that can be shortened. I think the overall finding that upwelling plays a considerable role in surface ocean DIC distribution stays even with the detailed calculations.

We tried our best to improve the readability in previous versions, in response to previous reviewer comments, and also to make the MS as concise as possible. We were able to trim a few additional sentences from the present version. For instance, we exclude some details in Section 4.4 which are not central to the narrative of the paper. Some sentences/paragraphs were revised throughout the manuscript as well.

[revised manuscript text omitted]